# DNA circles promote yeast ageing in part through stimulating the reorganization of nuclear pore complexes

Anne C Meinema[1†‡], Anna Marzelliusardottir[1†], Mihailo Mirkovic[1†], Théo Aspert[2], Sung Sik Lee[1,3], Gilles Charvin[2], Yves Barral[1*]

[1]Institute of Biochemistry, Department of Biology, ETH Zürich, Zürich, Switzerland; [2]Institute of Genetics and Molecular and Cellular Biology, Illkirch, France; [3]Scientific Center for Optical and Electron Microscopy, ETH Zürich, ETH Zürich, Switzerland

**Abstract** The nuclear pore complex (NPC) mediates nearly all exchanges between nucleus and cytoplasm, and in many species, it changes composition as the organism ages. However, how these changes arise and whether they contribute themselves to ageing is poorly understood. We show that SAGA-dependent attachment of DNA circles to NPCs in replicatively ageing yeast cells causes NPCs to lose their nuclear basket and cytoplasmic complexes. These NPCs were not recognized as defective by the NPC quality control machinery (SINC) and not targeted by ESCRTs. They interacted normally or more effectively with protein import and export factors but specifically lost mRNA export factors. Acetylation of Nup60 drove the displacement of basket and cytoplasmic complexes from circle-bound NPCs. Mutations preventing this remodeling extended the replicative lifespan of the cells. Thus, our data suggest that the anchorage of accumulating circles locks NPCs in a specialized state and that this process is intrinsically linked to the mechanisms by which ERCs promote ageing.

*For correspondence:
yves.barral@bc.biol.ethz.ch

†These authors contributed equally to this work

Present address: ‡Neogene Therapeutics, Amsterdam, Netherlands

Competing interest: The authors declare that no competing interests exist.

## Editor's evaluation

This interesting study examines a potential relationship between the tethering of extrachromosomal DNA (ecDNA) to the nuclear pore complex (NPC) and its role in aging; a model is proposed whereby the nuclear basket is displaced by ecDNA anchoring, which leads to a broader remodeling of the NPC that is distinct from NPC damage. This idea is conceptually novel and will represent an important advance, although some more support for the conclusions is still needed.

## Introduction

From yeast to mammals, nuclear pore complexes (NPCs), which mediate transport of cargos between nucleus and cytoplasm, undergo substantial changes during ageing (*Rempel et al., 2020*). In postmitotic cells such as neurons, core NPC components do not turnover, tend to become oxidized over time and progressively lose functionality (*Savas et al., 2012*; *Toyama et al., 2013*). Accordingly, in the neurons of old rats NPCs become increasingly leaky with age, affecting the proper retention of nucleoplasmic proteins in the nucleus (*D'Angelo et al., 2009*). On the opposite, yeast NPCs are targeted by a quality control machinery that removes damaged or misassembled NPCs (*Webster et al., 2014*; *Webster et al., 2016*). Accordingly, no oxidized NPCs are observed in replicatively ageing yeast cells (*Rempel et al., 2019*). Still, proteomics analysis of the old yeast cells indicate that they progressively change composition, losing their nuclear basket and cytoplasmic complexes (*Janssens et al., 2015*; *Rempel et al., 2019*). Similar changes are observed in the liver of ageing rats (*Ori et al., 2015*;

*Rempel et al., 2020*). However, little is known about how age drives these changes and whether this process is simply a consequence or contributes to ageing.

The replicative ageing of the budding yeast *Saccharomyces cerevisiae* is driven by its asymmetric mode of cell division, during which the larger mother cells bud daughters off their surface. These mother cells retain and accumulate diverse ageing factors (*Denoth Lippuner et al., 2014*; *Mortimer and Johnston, 1959*). Therefore, after generating 20–30 daughter cells they stop dividing and ultimately die. Retention of the ageing factors in the mother cells resets age and lifespan potential of their daughters.

Extra-chromosomal DNA circles are prominent ageing factors at least in yeast (*Denoth-Lippuner et al., 2014*; *Morlot et al., 2019*; *Sinclair and Guarente, 1997*). These circles are generated in virtually all mother cells at some point in their lifespan through excision of chromosomal fragments (*Moller et al., 2018*; *Morlot et al., 2019*). Due to its repeated nature and high copy number, the rDNA locus is the primary source of such circles, called extrachromosomal rDNA circles (ERCs). Non-chromosomal DNA circles are tightly retained in the mother during cell division (*Shcheprova et al., 2008*). Circles that contain a replication origin, like ERCs, replicate once per division cycle and accumulate exponentially in ageing mother cells over time. These cells may then contain up to a thousand ERCs at the end of their life, which represents about as much DNA as the rest of the genome (*Denoth-Lippuner et al., 2014*; *Morlot et al., 2019*). Several lines of evidence establish that ERC accumulation promotes cellular ageing. Mutants that form ERCs at a reduced rate, such as the *fob1Δ* mutant cells, show an extended replicative lifespan (RLS; *Defossez et al., 1999*). In reverse, cells in which recombination in the rDNA is derepressed, such as cells lacking the sirtuin Sir2, show a higher rate of ERC formation, accumulate them faster and are short-lived (*Kaeberlein et al., 1999*). Finally, artificially introducing an ERC (*Sinclair and Guarente, 1997*) or any other replicating DNA circle in young cells (*Denoth-Lippuner et al., 2014*; *Falcón and Aris, 2003*) causes their premature ageing. However, how DNA circles promote ageing is not known.

Yeast cells undergo a closed mitosis, the nuclear envelope and NPCs remaining intact through mitosis. Tight retention of DNA circles in the mother cell depends on their anchorage to the nuclear periphery, through NPCs (*Denoth-Lippuner et al., 2014*). Anchorage is mediated by SAGA, a large multi-subunit complex harboring acetyl-transferase activity (provided by Gcn5) and which binds both chromatin (*Durand et al., 2014*; *Wang et al., 2020*) and NPCs, via its subunit Sgf73 and the nucleoporin Nup1 (*Jani et al., 2014*; *Köhler et al., 2008*). ERC retention in wild type mother cells also causes the accumulation of the NPCs to which they are bound, such that ageing wild-type cells may contain up to 10-fold more NPCs than young ones (*Denoth-Lippuner et al., 2014*). Furthermore, mutations preventing ERC formation or accumulation dampen NPC accumulation with time and extend the replicative longevity of the cell. Thus, ERCs might promote ageing through their interaction with NPCs.

The yeast nuclear pore complex is a ~ 50 megadalton structure comprising repetitions of more than 30 different subunits, called nucleoporins (short Nups) (*Beck and Hurt, 2016*; *Kim et al., 2018*). The core of the NPC, also called the inner pore ring, is inserted in the nuclear envelope and stabilized by additional protein rings on both the cytoplasmic and nucleoplasmic sides of the pore (*Bui et al., 2013*). The center is filled by so-called FG-Nups, which both form a barrier to passive diffusion and facilitate the passage of transport cargos across the NPC (*Knockenhauer and Schwartz, 2016*; *Peters, 2009*). On their cytoplasmic face, NPCs are decorated by a complex of cytoplasmic Nups which have been also described as fibrils or filaments (*Fernandez-Martinez et al., 2016*; *Strambio-De-Castillia et al., 2010*; *Figure 1A*). On their nucleoplasmic side, they assemble the so-called nuclear basket formed mainly of the extended protein TPR, known as Mlp1 and Mlp2 in yeast (*Strambio-de-Castillia et al., 1999*), and anchored to NPCs by the protein Nup60. The basket mediates NPC-chromatin interaction (*Niepel et al., 2013*) and contributes to mRNA quality control and export (*Green et al., 2003*; *Vinciguerra et al., 2005*). Here, we investigate whether DNA circle anchorage affects NPC composition during replicative ageing and whether this limits cellular longevity. Our data indicate that circle binding to NPCs drives NPC remodeling and promotes cellular ageing through the accumulation of specialized NPCs and imbalance in nucleo-cytoplasmic exchange.

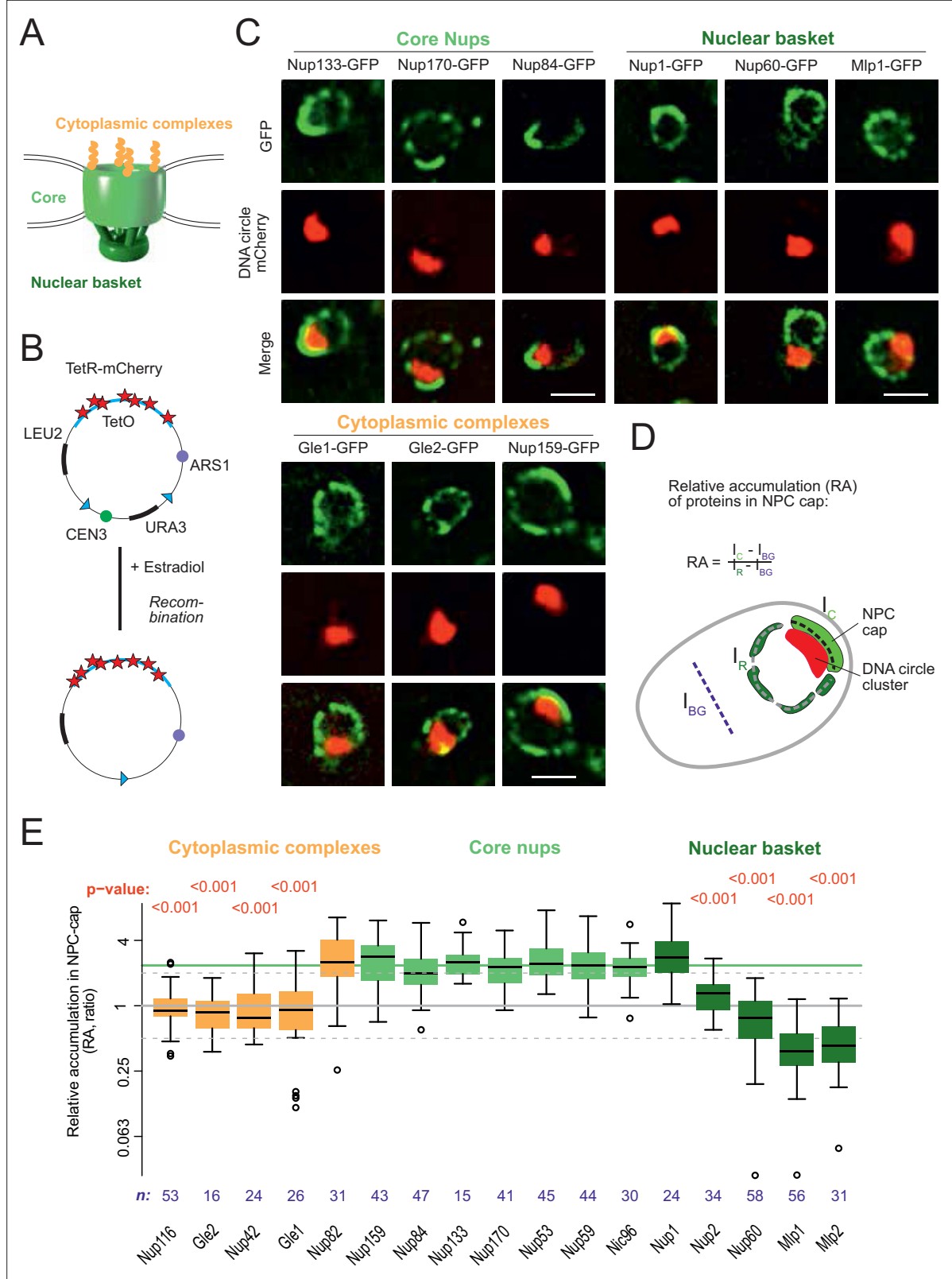

**Figure 1.** DNA circle anchored NPCs lack the peripheral subunits. (**A**) Cartoon of the NPC, showing the scaffold core, cytoplasmic complexes and nuclear basket. (**B**) Cartoon of an engineered DNA circle with an excisable centromere (CEN3) and TetR-mCherry decorated TetO repeats. The circle contains an autonomic replication sequence (ARS1) and selection markers (LEU2, URA3). (**C**) Fluorescent images of nuclei in yeast cells with accumulated DNA circles; the DNA circle clusters are labeled with TetR-NLS-mCherry (red) and different nucleoporins labeled with GFP (green). Scale bar is 2 μm.

*Figure 1 continued on next page*

*Figure 1 continued*

More images in *Figure 1—figure supplement 1*. (**D**) Cartoon exemplifying the quantification of protein accumulation around the DNA cluster. The NPC cap was selected by a line scan through the nuclear envelope adjacent to the DNA circle cluster. The relative accumulation (RA) is defined as the ratio of background corrected GFP intensity in the NPC cap ($I_c - I_{BG}$) over that in the rest of the nucleus ($I_R - I_{BG}$). (**E**) Quantification of GFP-labeled nucleoporin accumulation in the NPC cap localized at the DNA circle cluster. The relative fluorescence intensity in the NPC cap is plotted on a log2-scale. The median accumulation of the core nucleoporins is indicated (green line). p-Value stands for student's t-test between the specific nucleoporin and pooled data of all core nups together, no p-value is indicated if the difference is not significant; the sample size per strain is indicated (**n**).

The online version of this article includes the following source data and figure supplement(s) for figure 1:

**Source data 1.** Nups relative intensity in the cap versus the rest of the nuclear envelope.

**Figure supplement 1.** DNA circle anchored NPCs lack the peripheral subunits, Related to *Figure 1*.

## Results

### Circle-bound NPCs lack the nuclear basket

In order to study whether the anchorage of DNA circles to NPCs affects NPC composition, we took advantage of an engineered reporter DNA circle, which we have previously characterized (*Baldi et al., 2017*; *Denoth-Lippuner et al., 2014*; *Shcheprova et al., 2008*). This circle is generated in vivo by recombining it out of a mini-chromosome, using the R recombinase in a controlled manner (*Megee and Koshland, 1999*). It carries an array of 256 TetO repeats, and is, hence, visible as a fluorescent dot in cells expressing a fluorescently labeled TetR protein (*Figure 1B*, see Materials and methods and *Denoth-Lippuner et al., 2014*). This circle replicates in S-phase and remains confined in the mother cell upon cell division, where it accumulates through successive budding cycles. As they accumulate, the DNA circles cluster together to form a bright patch at the nuclear periphery. The attached NPCs accumulate at the nuclear envelope adjacent to the cluster (*Denoth-Lippuner et al., 2014*). Upon GFP-labeling of core Nups, these NPCs are visible as a cap of enhanced fluorescence density (*Figure 1C*, *Figure 1—figure supplement 1A*). In contrast, circle-free cells distribute their NPCs homogenously over the nuclear envelope (*Figure 1—figure supplement 1B*). In the few cells containing a cluster of circles, this system conveniently allows imaging circle-bound NPCs at the single cell level, and discriminating them from unbound NPCs in the remainder of the same nuclear envelope.

Using this system, we quantified the local enrichment of single Nups in the NPC cap compared to the rest of the nuclear envelope. We tagged 17 Nups with GFP, representative of different NPC subcomplexes, in cells where the DNA circle cluster was labelled in red (TetR-mCherry; *Figure 1B and C*). In these cells, all the stable core Nups, i.e., the scaffold components (outer ring: Nup84, Nup133; inner ring: Nup170, Nic96) and the components of the transport channel (FG-Nups: Nup53, Nup59) accumulated in the cap to similar extents. Quantification of their fluorescence indicated a 2.4-fold enrichment (median), compared to their localization elsewhere in the nuclear envelope (*Figure 1D and E*). All tested core Nups showed the same enrichment in the cap, indicating that the DNA circles bind intact NPC cores.

In striking contrast to core Nups, most peripheral subunits on both sides of the NPC did not accumulate in the cap. Four out of the five components of the nuclear basket and four out of the six components of the cytoplasmic complexes were present only at reduced levels or excluded from the cap (*Figure 1C and E*). Only Nup1, which appears to be more stably associated to NPCs than the other basket components (*Denoth-Lippuner et al., 2014*), and Nup83 and its partner Nup159, which form the docking site for the cytoplasmic complexes at the cytoplasmic side of NPCs, accumulated together with the core NPC. The basket component Nup2, and the cytoplasmic subunits Nup116, Nup42, Gle1, and Gle2 were present at reduced levels in the cap compared to core Nups. The basket components Mlp1, Mlp2 and Nup60 seemed literally excluded from the cap (*Figure 1E*). Altogether, these quantifications indicate that the circle-bound NPCs specifically lacked their nuclear basket and cytoplasmic complexes, compared to circle-free NPCs. Thus, they showed an altered stoichiometry very reminiscent of that observed for the NPCs of old yeast mother cells, as estimated by Mass-spectrometry (*Janssens et al., 2015*; *Rempel et al., 2019*).

## NPCs of wild-type cells aged under physiological conditions also lack basket and cytoplasmic complexes

Accumulating ERCs are rather dispersed throughout the nuclear periphery and form clusters only episodically, such that NPC caps are less prominently observed in old cells (see for example *Morlot et al., 2019*). Clustering of the reporter circle might be stabilized by the fluorescent label (mCherry still has a low affinity for itself). In order to study if NPCs change composition under physiological ageing conditions, we co-labeled pairs of Nups with distinct fluorophores and characterized their co-incorporation into NPCs by analyzing the spatial correlation of their fluorescence in the nuclear periphery of young and aged cells (*Figure 2A*). Indeed, the signals of the two labeled Nups should correlate well with each other as long as both colocalize to NPCs, and poorly if any one of them is displaced (*Figure 2A*).

In order to acquire images of old mother cells, the labelled cells were loaded on a microfluidic chip (*Jo et al., 2015*) and imaged for 26 hr (21 divisions on average). A continuous flow of fresh medium provided nutrients to the trapped mother cells and flushed the daughters out, allowing continuous imaging of the cells. Bright field images were taken every 15 min to monitor budding events and record the replicative age of each cell. Fluorescence images were acquired after 2 and 26 hr (i.e. on average 2 and 21 divisions, respectively, *Figure 2B*). Colocalization of the labeled Nups was then quantified (*Figure 2C*).

Analysis of Pearson correlation between the signals of core Nups (Nup159-mCherry and Nup170-GFP), showed that it was similarly high in young and old mother cells (average correlation = 0.78 ± 0.03, n = 33 and 0.79 ± 0.02, n = 30, respectively; *Figure 2C*). Thus, these two proteins indeed remain stable at NPCs of old cells. In contrast, the basket proteins Nup60 and Mlp1 and the cytoplasmic protein Gle1 (fused to GFP) colocalized less extensively with the core NPC (Nup159-mCherry) already in young cells (average correlation = 0.68 ± .04, n = 32; 0.61 ± 0.04, n = 31 and 0.62 ± 0.05; n = 32 respectively), consistent with these proteins associating more transiently with NPCs (*Denning et al., 2001*; *Denoth-Lippuner et al., 2014*; *Dilworth et al., 2001*; *Niepel et al., 2013*). Moreover, the correlation substantially dropped in old cells (average correlation 0.35 ± 0.05, n = 33; 0.31 ± 0.06, n = 37 and 0.24 ± 0.06, n = 35 respectively; p < 10$^{-4}$; *Figure 2C*). A similar drop of correlation was observed when comparing the signals of Mlp1-GFP and Nup84-mCherry (*Figure 2D and E*), indicating that it was independent of the core Nup used as reference. Thus, in yeast mother cells that underwent unperturbed ageing, a substantial fraction of the NPCs lacks a basket and cytoplasmic complexes, in line with mass-spectrometry data from old yeast cells (*Janssens et al., 2015*; *Rempel et al., 2019*).

## ERCs mediate the accumulation of basket-less NPCs in old mother cells

To determine whether ERC accumulation drives the re-composition of old cells' NPCs, we next asked whether preventing ERC accumulation restored the stoichiometry of NPCs. Thus, we deleted the *FOB1* gene and monitored Nup60 and Mlp1 localization relative to core Nups upon ageing (*Figure 3A*). Strikingly, unlike wild type cells of the same age, the aged *fob1Δ* mutant cells did not show any significant dissociation of the basket from NPCs after 22–24 budding cycles compared to young wild type and young *fob1Δ* mutant cells (*Figure 3A*). Moreover, deleting the *SGF73* gene, which docks SAGA to NPCs and mediates circle anchorage to nuclear pores (*Denoth-Lippuner et al., 2014*), also restored the localization of the basket to NPCs in aged cells (*Figure 3B*). Thus, ERC presence, attachment and SAGA function are required for the basket to be displaced from NPCs in old cells.

As ERC anchorage promotes accumulation of basket-less NPCs in old wild type mother cells, we reasoned that their daughters, which do not inherit ERCs, should rapidly recover the proper localization of the basket proteins. Indeed, when comparing signal correlation between the basket proteins Nup60 or Mlp1, with the core Nup, Nup159, colocalization between basket and core NPCs was restored in the rejuvenated daughter cells of old mothers, to the level observed in both young mothers and their daughters (*Figure 3C and D*). We conclude that the formation of ERCs and their attachment to NPCs drives the accumulation of basket-less NPCs in old cells.

## The cell does not recognize circle-bound NPCs as defective

Several studies have indicated that NPCs can deteriorate with time and established the existence of a machinery dedicated to removing damaged or misassembled NPCs in yeast. This quality control system, called Storage of Improperly assembled NPCs Compartment (SINC), recruits the ESCRT III

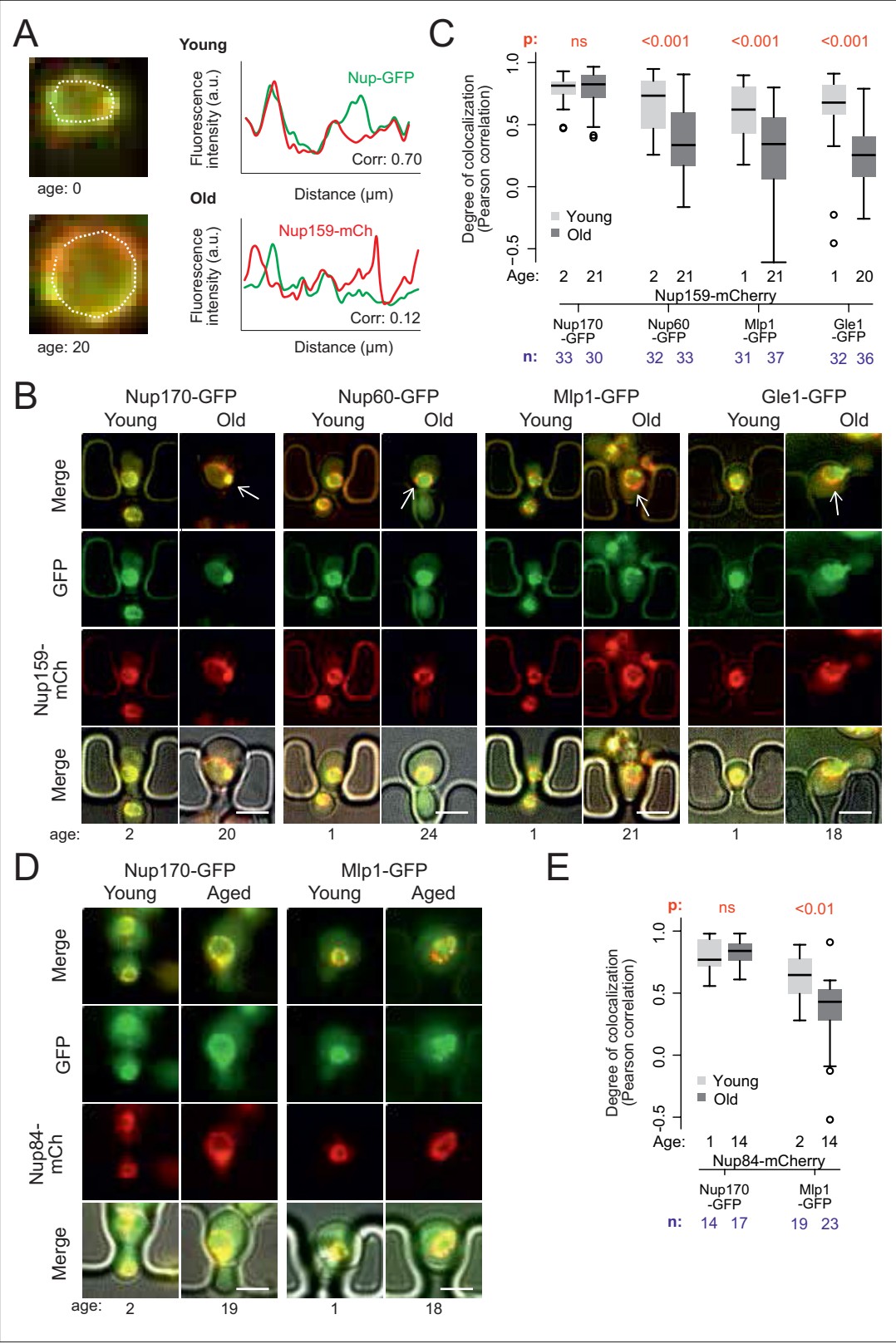

**Figure 2.** Peripheral subunits of NPC displaced in wild-type aged cells. (**A**) To quantify the degree of nucleoporin colocalization, a line was drawn through the nuclear envelope in a focal image of a young (top) and aged (bottom) nucleus of cells co-expressing GFP-tagged Nup GFP (e.g. Nup60-GFP) and a mCherry-tagged Nup as a reference (e.g. Nup159-mCherry, see Materials and methods). The Pearson correlation between the two intensity profiles is calculated, and used as a measure for nucleoporin colocalization. (**B**) Fluorescent images of young and old cells in the yeast ageing chip,

*Figure 2 continued on next page*

*Figure 2 continued*

with nucleoporins labeled with GFP (green) and the reference Nup159 with mCherry (red). The age of each cell is indicated. Scale bars are 5 μm. (**C**) Quantification of the degree of colocalization between target and reference nucleoporin is plotted for young and old wild type cells. The p-value stands for the student's t-test between young and old cells. The sample size (**n**) and the median age is indicated. (**D**) Fluorescent images of young and aged cells in the yeast ageing chip, as in B, but with Nup84-mCherry as a reference nucleoporin. Scale bars are 5 μm. (**E**) Quantification of the co-localization between nucleoporins in young and old cells, as in C, but with Nup84-mCherry as a reference.

The online version of this article includes the following source data for figure 2:

**Source data 1.** Correlation data Core/Peripheral Nups according to age.

machinery, including its component Snf7, the adaptor protein Chm7 and the AAA-ATPase Vps4, to defective NPCs and mediates their removal from the nuclear envelope (*Rempel et al., 2019*; *Webster et al., 2016*). Thus, we next asked whether the attachment of DNA circles and basket loss are recognized as damaged NPCs by the cell. We monitored whether circle-bound NPCs recruit Chm7, Vps4, or Snf7 more frequently than bulk NPCs. We first recorded the localization of these proteins, tagged with GFP, in cells loaded with DNA circles and asked whether they decorated the NPC cap (*Figure 4A–B*). In young cells, GFP-labelled, endogenous Vps4 and Snf7 are distributed in several larger dots per cell, with Snf7 forming a handful of additional smaller dots. In cells that accumulated the reporter DNA circle and assembled a cluster, none of the larger or smaller Vps4 and Snf7 dots appeared to localize particularly frequently to the periphery of the clusters or to colocalize with them (*Figure 4A*). Quantification of the GFP fluorescence emitted by either of these labeled proteins at the nuclear periphery surrounding or away from the cluster (depicted in *Figure 4B*, as in *Figure 1D*) indicated that neither Vps4 nor Snf7 accumulated near the circle cluster more than anywhere else in the nuclear envelope (*Figure 4C*). Similarly, Chm7-GFP, an SINC marker (*Webster et al., 2016*), which shows a more diffuse localization but still forms several brighter dots per cell, did not accumulate more frequently around the circle cluster than elsewhere (*Figure 4—figure supplement 1*.). Thus, we found no evidence that the the NPCs associated with the reporter DNA circle would be particularly targeted to the SINC or particularly recruit the ESCRT system, indicating circle-bound NPCs are not recognized as being defective by the cell.

To challenge this notion, we then investigated whether ERC accumulation would lead to the formation of NPCs that are recognized as defective in old cells. To this end, we labeled with mCherry the endogenous protein Net1, which associates with the rDNA locus both on the chromosome and on ERCs (*Morlot et al., 2019*; *Neurohr et al., 2018*) and monitored its accumulation in cells ageing in our microfluidics chip (*Figure 4D*). In the same cells, we labeled either Vps4 or Snf7 with GFP to investigate whether these ESCRTs subunits decorate the rDNA foci accumulating with age. Indeed, if the NPCs associated with ERCs are targeted by the ESCRT system, this should be reflected by some recruitment of Vps4 and Snf7 to the vicinity of Net1 decorated ERCs. As described, the amount of Net1-decorated material indeed increased more than 10-fold in the nucleus of the old mother cells compared to young ones, consistent with these cells accumulating ERCs with age (*Figure 4D*). In these cells, the signal for both Vps4 and Snf7 increased as well and became somewhat more diffuse. However, we did not find any accumulation of these proteins immediately around or at the Net1 clusters compared to the rest of the cell (*Figure 4E*). Thus, neither the NPCs associated with the reporter circle in young cells nor those associated with the ERCs in old cells are recognized as defective by the cell.

## Basket loss does not correlate with the age of NPCs

Since the anchorage of a DNA circle to an NPC causes its retention in the mother cell, we next wondered whether this could cause the progressive accumulation of older NPCs in aged mother cells and in turn, the loss of their basket. To test this possibility, we measured the relative age of NPCs in old mother cells and their daughters using a tandem fluorescence timer, consisting of a mCherry (mCh) and superfolder GFP (sfGFP) fusion protein (*Khmelinskii et al., 2012*). Due to different maturation kinetics between the two fluorophores, a newly synthesized protein appears first in the green channel before acquiring the red fluorescence over time. As the turnover rate of Nup170 is very low in NPCs (*D'Angelo et al., 2009*; *Hakhverdyan et al., 2021*) older pores with tagged Nup170 are expected to emit more red fluorescence than green in comparison to newly assembled pores. To see if old mothers are enriched in red-shifted old pores, we loaded the cells expressing Nup170-mCh-sfGFP on the microfluidic chip and imaged them as above (*Figure 2B*) as the cells aged (*Jo et al., 2015*).

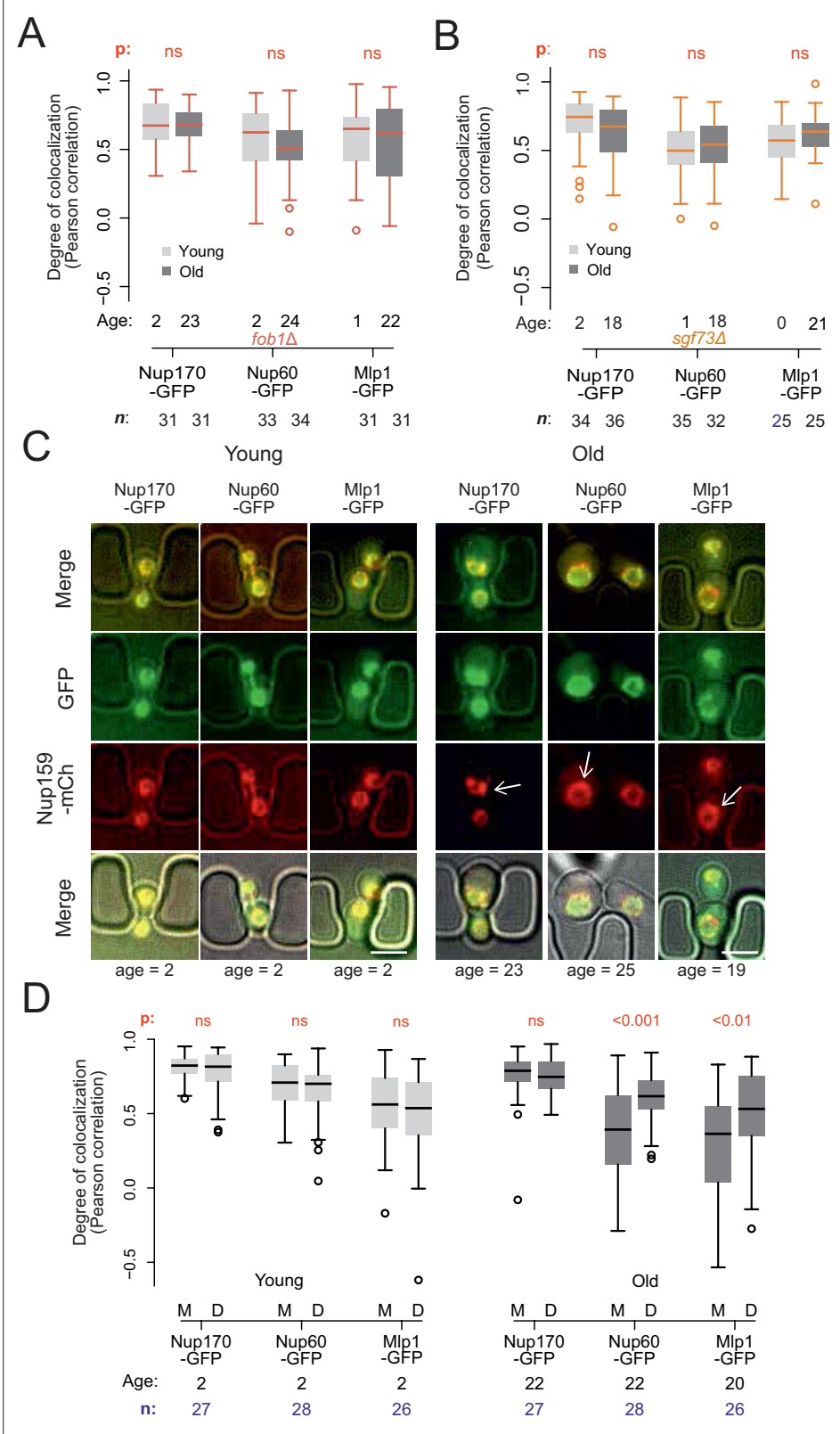

**Figure 3.** Endogenous DNA circles drive nuclear basket displacement in wild-type aged cells. (**A**) Quantification of the degree of colocalization between target and reference nucleoporin is plotted for young and old wild-type, as was done in *Figure 2C*, but in *fob1Δ* cells. The sample size (**n**) and the median age is indicated. The p-value stands for the student's t-test between young and old cells. (**B**) Same as A, but in *sgf73Δ* cells. (**C**) Fluorescent

*Figure 3 continued on next page*

*Figure 3 continued*

images of young and old mitotic cells in the yeast ageing chip. Scale bars are 5 μm. (**D**) Same as in *Figure 2C*, but comparing nucleoporin colocalization in young and old mother (**M**) with their corresponding daughter (**D**) cell. The p-value stands for the student's t-test between mother and daughter cell. The sample size (**n**) and the median age is indicated.

The online version of this article includes the following source data for figure 3:

**Source data 1.** Correlation data Core/Basket Nups in fob1Δ mutant cells.

**Source data 2.** Mother/Daughter distribution Core/Basket Nups with age.

The fluorescence channels were recorded after 2 and 26 hr (*Figure 4F*). Quantification of the signal (*Figure 4G*) indicated a tendency for young cells to put slightly more red shifted NPCs in the bud than in the mother cell (not statistically significant, n = 31 mother-bud pairs), as reported (*Khmelinskii et al., 2012*). However, over time we also observed a highly significant increase of the red to green signal ratio in old compared to young mother cells (n = 29 old cells, p < 0.001), indicating that old cells might actually accumulate old pores over time. Strikingly, the NPCs of their daughter cells were not significantly less red-shifted in average (*Figure 4G*). Thus, although old mother cells might accumulate older NPCs, the composition difference of these NPCs between mother and daughter cells is not driven by their age.

## Circle-anchorage promotes very specific changes in NPC-karyopherin interactions

We next characterized the effect of DNA circle anchorage on NPC functionality. A broad variety of transport factors, generically called karyopherins, transiently localize to nuclear pores in young and healthy cells to mediate nucleo-cytoplasmic exchanges (*Derrer et al., 2019*; *Kumar et al., 2002*). Any effect of NPC remodeling on the localization of these factors may reflect changes in their dynamics within NPCs and hence, on NPC functionality. Therefore, we characterized how circle anchorage affected the recruitment of transport factors and other associated proteins to NPCs. We labeled a broad panel of these factors with GFP and quantified their nuclear localization in respect to the DNA circle cluster, as in *Figure 1*. Only the factors showing a clear (transient) localization to the nuclear envelope of young wild-type cells were characterized. Consistent with circle-bound NPCs lacking a basket, the two basket-associated proteins Esc1 and Ulp1 (both involved in telomeric silencing and mRNA surveillance; *Bonnet et al., 2015*) were excluded from NPC caps (*Figure 5A and B*). In striking contrast, none of the 10 importins tested were displaced from the cap (*Figure 5B*). Likewise, the exportins Xpo1 (Crm1), Msn5 and Los1, which ensure the export of proteins and tRNAs, accumulated at the cap to similar extent as the core Nups. Since nearly all these proteins accumulated to the same extent as core NPCs in the cap compared to elsewhere in the nuclear envelope, we concluded that basket-less pores interact with them with similar dynamics as bulk NPCs.

However, two importins were significantly enriched in the cap compared to bulk NPCs, namely Srp1/Kap60 and Kap123. To test whether this reflected some stiffening of circle-bound NPCs, which could slow down the passage and increase the retention of these importins and their relevant cargos, we used fluorescence loss in photobleaching (FLIP) to ask whether the shuttling of these proteins at NPCs was altered upon circle-binding. We C-terminally labeled the endogenous proteins Kap60 and Kap123 with GFP individually in cells carrying the reporter circle and asked whether fluorescence decay was slower at the labelled NPCs adjacent to the cluster of DNA circles (labelled with TetR-mCherry) than elsewhere at the nuclear envelope upon constant bleaching in the center of the nucleus (*Figure 6A*). Quantification of the fluorescence signal showed that both proteins decayed with identical kinetics at the cap-NPCs as elsewhere in the nuclear envelope (*Figure 6B*). Thus, importins' interaction with circle-bound and circle-free NPCs followed undistinguishable kinetics. Somehow the absence of the basket or the presence of an ERC increases the frequency with which Kap60 and Kap123 interact with NPCs but not the time they spend in them.

Consistent with circle-bound NPCs being fully functional for nuclear import and not leaky (as already seen in *Morlot et al., 2019*; *Rempel et al., 2019*), old cells accumulated the import reporter protein NLS-3xGFP in the nucleus at least as tightly as young cells (*Figure 6C–E*). In fact, the nucleo/cytoplasmic ratio of fluorescence show that old cells accumulate NLS-3xGFP more tightly in the nucleus

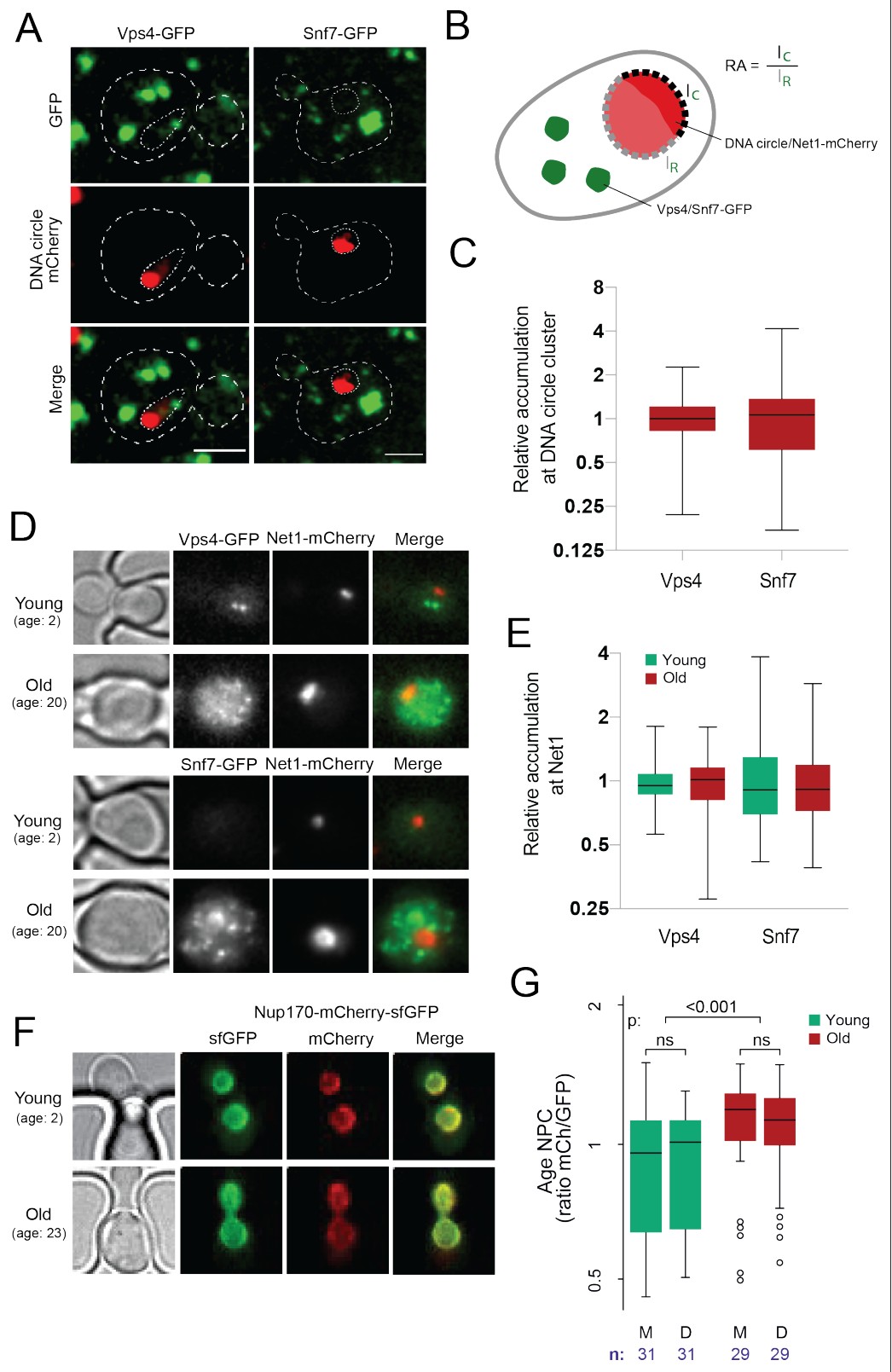

**Figure 4.** DNA circle anchored NPCs are not recognized as defective. (**A**) Cells with accumulated DNA circles labeled with TetR-NLS-mCherry and ESCRT-complex components Vps4 or Snf7 tagged with GFP. (**B**) Cartoon exemplifying the quantification of Vps4/Snf7 accumulation around the DNA circle cluster/Net1-mCherry focus. A line scan was drawn over the nuclear envelope adjacent to the DNA circle cluster/Net1-mCherry focus and another

*Figure 4 continued*

one over the rest of the nuclear envelope, in a single focal plane. The intensity of Vps4-GFP or Snf7-GFP was measured in each region. The relative accumulation (RA) is defined as the ratio of GFP intensity around the DNA circle cluster/Net1-mCherry focus ($I_C$) over that in the rest of the nuclear periphery ($I_R$). (**C**) Quantification of relative accumulation of Vps4-GFP and Snf7-GFP at the DNA circle cluster (n = 60 cells each). (**D**) Young and old cells in the yeast ageing chip. Net1 is tagged with mCherry and ESCRT-complex components Vps4 or Snf7 with GFP. (**E**) Quantification of the relative accumulation of Vps4 and Snf7 at Net1-mCherry foci in young and old cells (n = 50 cells each). (**F**) Young and old cells in the yeast ageing chip. Nup170 is tagged with a fluorescent timer (Nup170-mCherry-sfGFP). (**G**) Quantification of the relative age of the NPC plotted as the intensity ratio between mCherry and GFP of Nup170 tagged with a fluorescent timer (Nup170-mCherry-sfGFP), measured in young and old mother (**M**) versus daughter cells (**D**). Sample size (**n**) is indicated.

The online version of this article includes the following source data and figure supplement(s) for figure 4:

**Source data 1.** ESCRT/NPC colocalization and NPC age data.

**Figure supplement 1.** Chm7 does not accumulate at the DNA circle cluster.

---

than young ones (*Figure 6E*). We conclude that the NPCs of old yeast cells, including ERC-associated NPCs, are fully functional in the nuclear import and retention of proteins.

Interestingly however, our quantifications indicated that 5 exportins were specifically depleted from cap NPCs (*Figure 5B*), indicating that the interaction of these proteins with core NPCs was substantially decreased. Interestingly, all these exportins are involved in mRNA export. Thus, all seven proteins that we find to be excluded from the cap are involved in this process (*Bonnet et al., 2015*; *Iglesias et al., 2010*; *Stewart, 2010*). Consistent with the circle-bound NPCs being defective at mRNA export, they are also deprived of the TREX-2 complex, as judged by assaying the localization of its Sac3 subunit (*Figure 5B*). Together, these observations indicate that circle-bound NPCs are not fundamentally defective but may rather show a very specific decrease in mRNA export activity and an increased capacity for protein import.

## Nucleoporin acetylation promotes NPC remodeling

Since the SAGA complex accumulates on DNA circles and mediates their attachment to NPCs (*Denoth-Lippuner et al., 2014*), and given the fact that circle-bound NPCs are not recognized by the cell as defective, we finally wondered whether basket removal from circle-bound NPCs could be more akin to a regulatory event driven by SAGA than a defect. Particularly, we considered the possibility that SAGA-dependent acetylation of basket components promotes their detachment from circle-associated NPCs. Indeed, three of the four basket proteins displaced from circle-bound NPCs are acetylated in vivo, at least in part through SAGA (*Downey et al., 2015*; *Henriksen et al., 2012*). Particularly, Nup60, which anchors the Mlp proteins to NPCs (*Feuerbach et al., 2002*) is prominently acetylated on lysine 467 in a Gcn5-dependent manner (*Choudhary et al., 2009*; *Choudhary et al., 2014*). Thus, we asked whether specifically preventing its acetylation may circumvent its displacement from circle-bound NPCs. We substituted lysine-467 by arginine, a similarly charged but non-acetylatable residue, and imaged the distribution of the protein (as in *Figure 1C and E*). Strikingly, not only did Nup60-K467R accumulate in the cap nearly as much as core Nups, but the localization of the cytoplasmic complexes (Nup116, Nup42, Gle1, and Gle2) to cap NPCs was also restored (*Figure 7A and B*). In contrast, the basket protein Mlp1 remained displaced. Thus, acetylation of Nup60 on K467 upon circle attachment displaces it from NPCs and somehow also subsequently dissociates the cytoplasmic complexes. Since Nup60 mediates the anchorage of Mlps to NPCs (*Feuerbach et al., 2002*), Nup60 acetylation alone explains the detachment of the rest of the basket. However, our data indicate that additional events re-enforce the displacement of Mlp1/2, in addition to Nup60 acetylation. These events might involve their own acetylation or that of Nup2.

## Basket displacement promotes ageing

Finally, we wondered whether Nup60 acetylation and displacement from circle-bound NPCs contributes to the effect of ERCs in cellular ageing. We took advantage of an improved microfluidic chip, which more efficiently retained cells during their entire lifespan (*Figure 8A and B*; > 95% of the cells were kept until death, see methods and *Jacquel et al., 2021*) and asked whether preventing Nup60 acetylation had an effect on longevity. In this optimized setup, the wild type cells reproducibly showed

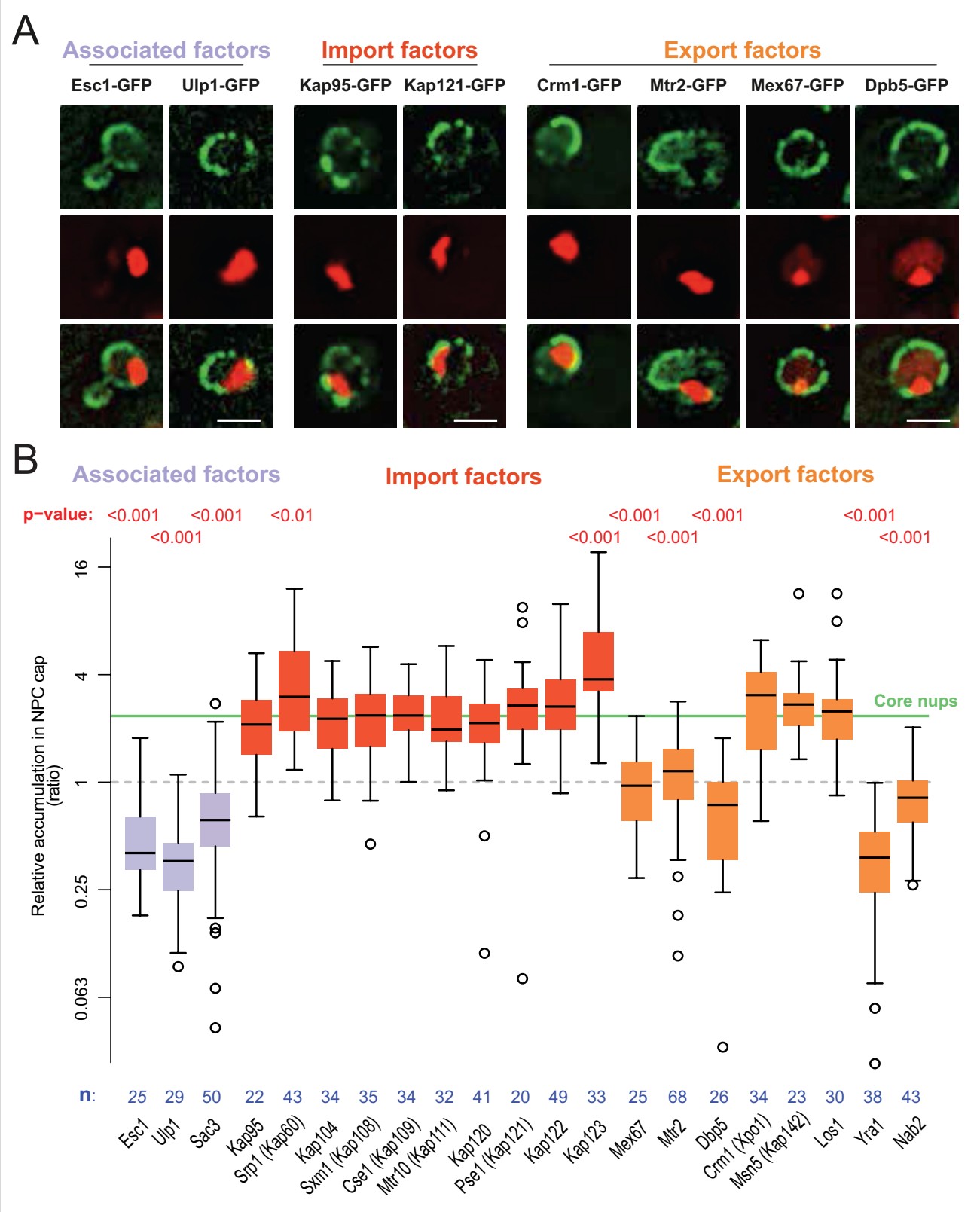

**Figure 5.** mRNA export and surveillance factors are displaced from circle-bound NPCs. (**A**) Fluorescent images of nuclei in aged yeast cells, where the DNA circle clusters are labeled with TetR-NLS-mCherry (red) and transport factors with GFP (green). Scale bar is 2 μm. (**B**) Quantification of factor recruitment in the NPC cap adjacent to the DNA circle cluster, as in *Figure 1E*. The protein accumulation is plotted for associated factors (purple),

*Figure 5 continued on next page*

*Figure 5 continued*

import factors (red), and export factors (orange), and is compared to accumulation of the pooled NPC core subunits NPC (green line, duplicated from *Figure 1E*), no p-value is indicated if the difference is not significant. The sample size (**n**) is indicated.

The online version of this article includes the following source data for figure 5:

**Source data 1.** Relative intensity of NPC associated factors in the cap versus the rest of the nuclear envelope.

a replicative life span of 18 generations (*Figure 8C*), well within the range of lifespans reported for wild-type cells measured on different microfluidic platforms (*Chen et al., 2017*; *Janssens and Veen-hoff, 2016*). These measurements established that mutant cells lacking the central basket component Nup60 aged rapidly (median life span of 13 divisions). In reverse, the *nup60-K467R* mutant cells showed a modest but significantly extended longevity (20 divisions, *Figure 8C*), despite the proteins Mlp1 and Mlp2 still being displaced from NPCs. Thus, we conclude that basket removal promotes ageing and that stimulation of this process by ERCs and SAGA limits the longevity of yeast cells.

## Discussion

Together, these studies make three main points. First, they not only confirm that ERCs and DNA circles in general attach to NPCs in ageing yeast cells, they establish that this in turn affects the organization and function of the NPCs retained in the mother cell. Thereby, anchorage to NPCs contributes to the mechanisms by which extrachromosomal DNA circles promote cellular ageing. Thus, circle-attachment to NPCs is a fundamental driver of ageing, at least in yeast.

Second, circle anchorage does not appear to damage NPCs but rather to activate a regulatory process that involves the compositional and functional reorganization of NPCs. Thus, circles might drive ageing through an imbalance in or overabundance of specialized NPCs.

Third, these observations underline the plasticity of NPCs and their capacity to remodel and specialize themselves in response to distinctive signals.

Altogether, we speculate that our observations reveal conserved regulatory properties of the nuclear pore and are broadly relevant for understanding ageing.

### NPC anchorage and mitotic partition of DNA circles

A growing body of evidence indicates that the asymmetric partition of DNA circles during yeast mitosis involves their attachment to NPCs and subsequent confinement into the mother cell by a lateral diffusion barrier (*Denoth-Lippuner et al., 2014*; *Clay et al., 2014*; *Ouellet and Barral, 2012*; *Shcheprova et al., 2008*). Indeed, abrogating this barrier, made of phytoceramide (*Clay et al., 2014*; *Megyeri et al., 2019*; *Prasad et al., 2020*), causes NPCs and DNA circles to leak to the bud, extending the mother cells lifespan (*Boettcher et al., 2012*; *Megyeri et al., 2019*; *Shcheprova et al., 2008*). Releasing DNA circles from their anchorage to NPCs by SAGA has the same effects (*Denoth-Lippuner et al., 2014*; *McCormick et al., 2014*). However, in apparent contradiction to this idea, previous studies have established that artificially binding the basket proteins Nup2 or Mlp1 (fused to TetR) to DNA circles releases them from retention in the mother cell (*Khmelinskii et al., 2011*), suggesting that tethering circles to NPCs bypasses rather than promotes their confinement in the mother cell. Nevertheless, that study and a subsequent one (*Denoth-Lippuner et al., 2014*) also indicated that tethering circles to core Nups did not impair their retention in the mother cell. Rather, it restored their retention when the endogenous, SAGA-dependent mechanism of anchorage was abrogated.

Here, we establish that at least in yeast the anchorage of DNA circles to NPCs directly underlies the reorganization of NPC characteristically observed in old yeast and liver cells, namely the loss of the basket and parts of the cytoplasmic complexes (*Janssens et al., 2015*; *Ori et al., 2015*; *Rempel et al., 2019*). Thus, our results provide an explanation for why anchorage of circles to basket proteins affects the retention of circles in the mother cell while anchorage to core NPCs does not. The absence of Mlp1 and Nup2 from circle-bound NPCs indicate that DNA circles and the nuclear basket exclude each other at NPCs. Thus, decorating a model DNA circle with basket proteins may cause both circle and basket to detach rather than to dock to NPCs, explaining their random segregation between mother and bud. This would account for all available data (*Denoth-Lippuner et al., 2014*; *Khmelinskii et al., 2011*; *Shcheprova et al., 2008*). Future studies will be needed to determine how the nuclear

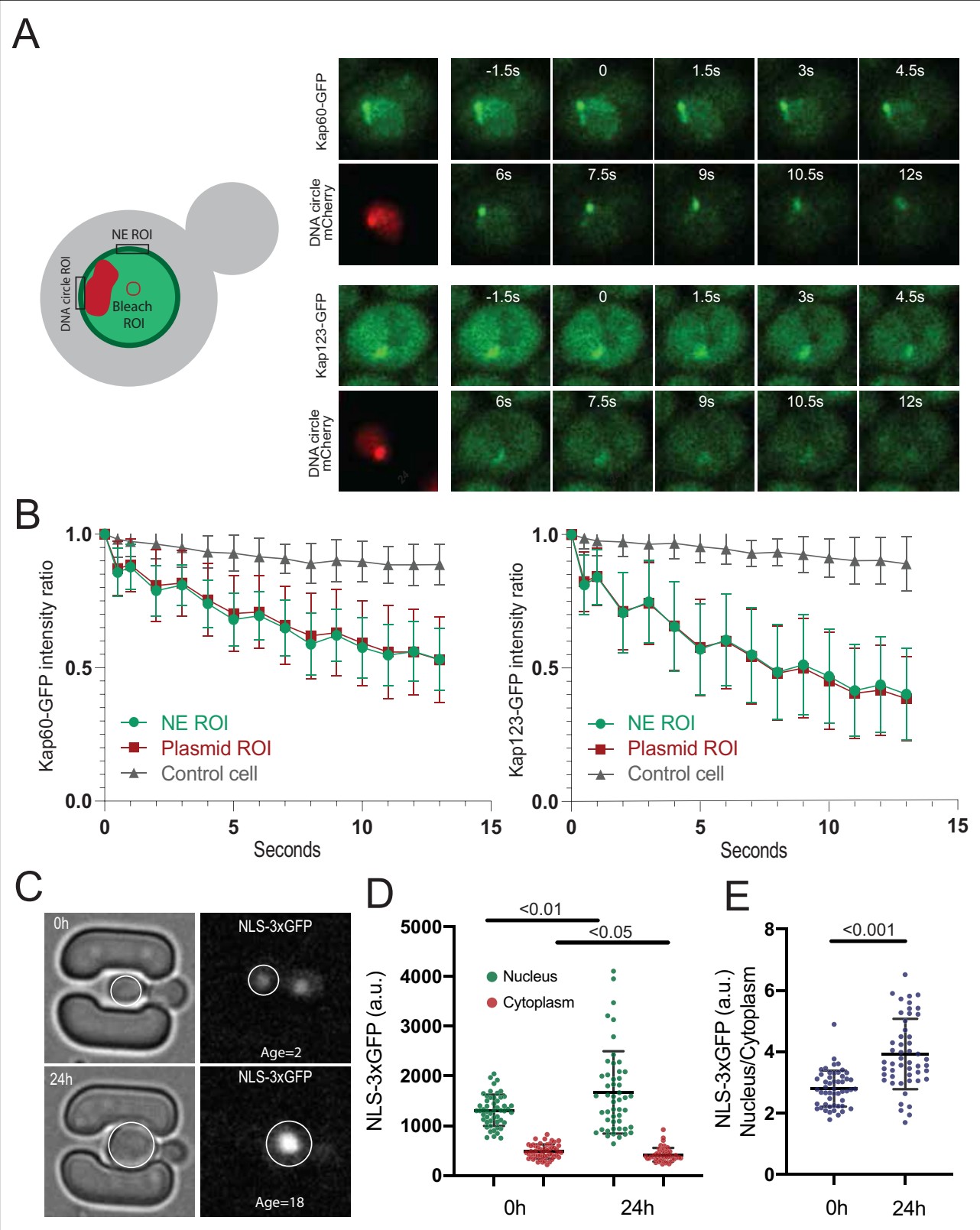

**Figure 6.** DNA circle bound NPCs are functional in the nuclear import and retention of proteins. (**A**) Fluorescent images from FLIP experiments on Kap60-GFP and Kap123-GFP with a DNA circle cluster (labeled with TetR-NLS-mCherry). A region within the nucleus was continuously bleached for 13 s and (**B**) the intensity kinetics of Kap60 and Kap123 were measured at the Nuclear Envelope (NE) at the DNA circle cluster and in the adjacent NE region, in a single focal plane (n = 33). (**C**) Example images from a microfluidics ageing experiment of NLS-3xGFP strain. Intensity of the GFP was measured

*Figure 6 continued on next page*

*Figure 6 continued*

in the nucleus and the cytoplasm of a single focal plane and quantifications are shown in (**D**) and (**E**) (n = 50 at each timepoint). Statistical significance assessed by unpaired t-test. Average replicative lifespan at 24 hr is ~18 divisions.

The online version of this article includes the following source data for figure 6:

**Source data 1.** Kap60-GFP and Kap123-GFP FLIP and NLS-3xGFP intensity in ageing cells.

## The NPCs of old cells

Next, our data provide concrete evidence that the basket-less NPCs accumulating in the mother cell qualify as legitimate ageing factors: Preventing NPCs from losing at least part of their basket and cytoplasmic complexes extends the longevity of the cell. Ageing factors are generally conceptualized as damaged components or other waste that the cell cannot properly eliminate. The fact that circle-bound NPCs lack the cytoplasmic complexes, some components of which are essential for viability (such as Gle1), and of some FG-Nups (Nup116, Nup42, and Nup60) could support the idea that they are simply defective. Circle attachment might damage the NPC or vice versa, circles might attach preferably to damaged NPCs.

Surprisingly, however, we could not obtain evidence for these NPCs being damaged. The NPCs of old yeast mother cells remain tight and promote the accumulation of nuclear proteins in the nucleus. The ERC and circle-bound NPCs are neither segregated to the SINC nor targeted by ESCRTs, which removes damaged or unproperly assembled NPCs, any more than other ones in the cell.

In addition, several data indicate that these NPCs were properly assembled to start with. The case of the cytoplasmic complex components Nup116, Nup82 and their partner Nup159 is emblematic in that respect. Indeed, Nup116 is essential for the recruitment of Nup82 and Nup159 to NPCs

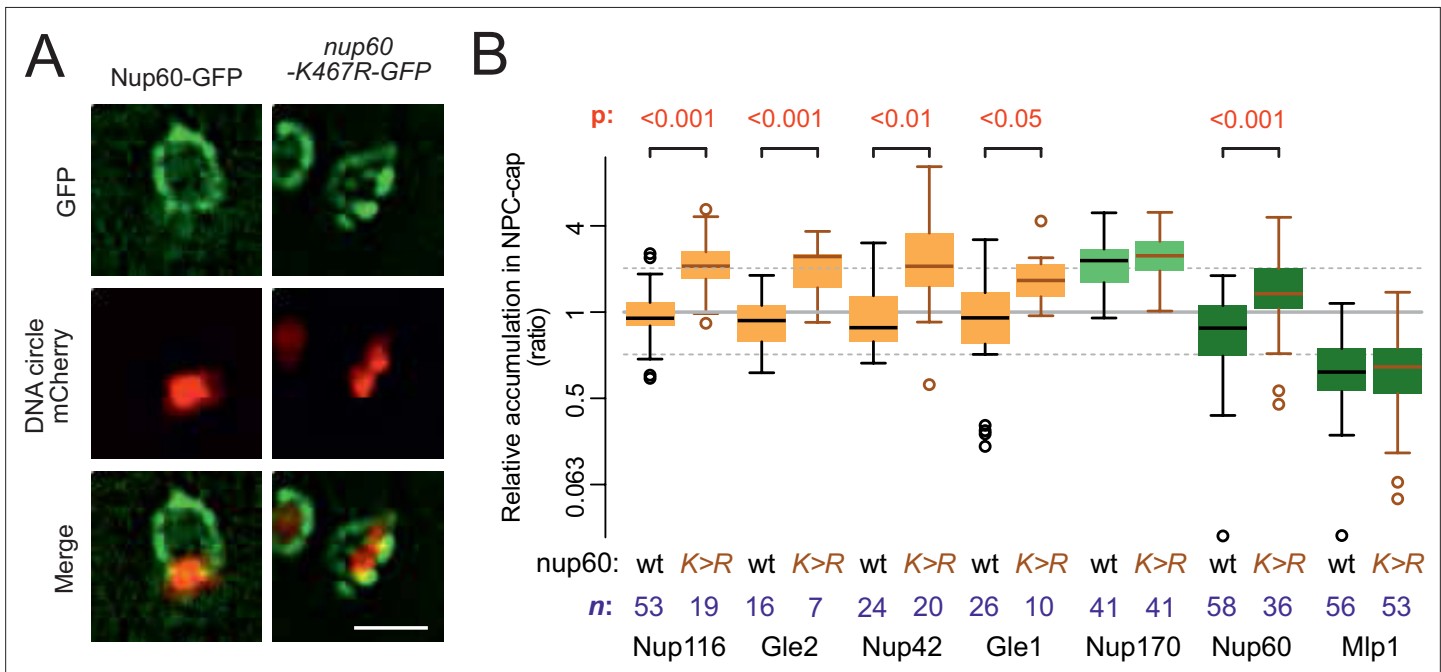

**Figure 7.** Nup60 acetylation mediates its removal from NPCs as well as that of the cytoplasmic complex upon circle anchorage. (**A**) Example images of nuclei in yeast cells with accumulated DNA circles and Nup60-GFP and nup60-K467R-GFP. The DNA circle clusters are labeled with TetR-NLS-mCherry (red). Scale bar is 2 μm. (**B**) Quantification of GFP-labeled nucleoporin accumulation in the cap, in nup60-K467R (brown lines), compared to wild-type (black lines) on a log2-scale, as in *Figure 1E*. Wild-type data is a copy from *Figure 1E*. The p-value stands for the student's t-test between accumulation ratio of a specific nucleoporin in wild-type and nup60-K467R, no p-value is indicated if the difference is not significant. The sample size (**n**) is indicated.

The online version of this article includes the following source data for figure 7:

**Source data 1.** Relative intensity of cytoplasmic complex and nuclear basket factors in the cap versus the rest of the nuclear envelope in nup60-K467R.

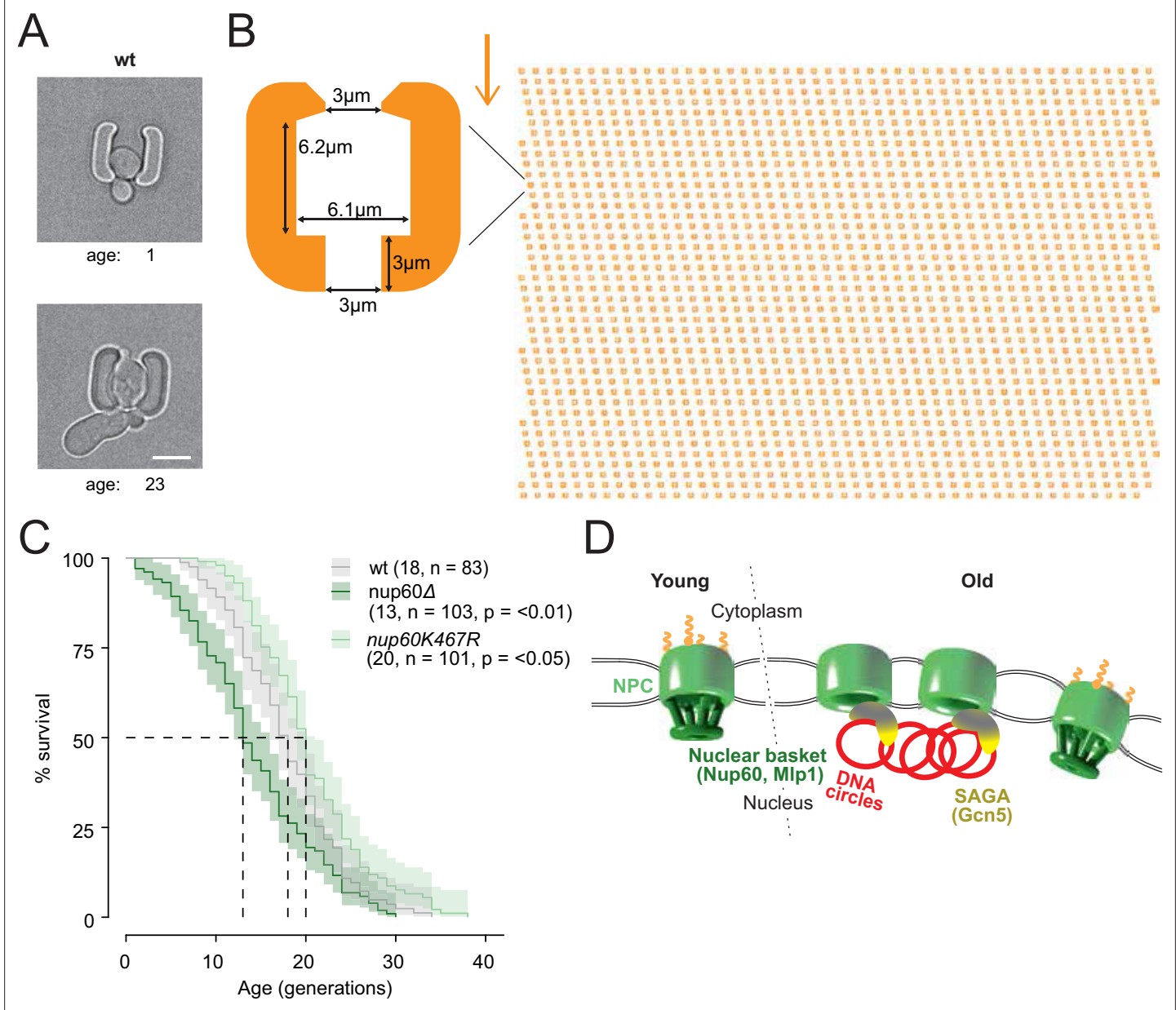

**Figure 8.** Basket displacement promotes ageing (**A**) Example images of the same young and old wild-type cell in a trap of the yeast ageing chip. The age of the cell is indicated, scale bar is 5 μm. (**B**) The design of the array with improved traps in the ageing chip. One single trap is highlighted to show its dimensions. The arrow shows the direction of the medium flow. (**C**) The lifespan curve for *nup60Δ* and *nup60-K467R* compared to a wild-type strain, plotted with 95% confidence interval limits. The p-value stands for Log-Rank test between *nup60Δ* or *nup60-K467R* with the wild-type strain. (**D**) Model depicting that DNA circle accumulation affects the organization of the NPC via a SAGA-mediated basket displacement.

The online version of this article includes the following source data for figure 8:

**Source data 1.** Replicative lifespan of wt, nup60del and nup60-K467R cells.

(***Allegretti et al., 2020***). Thus, the proper presence of Nup82 and Nup159 in circle-bound NPCs indicates that these NPCs must have originally contained Nup116, which they must have lost ulteriorly. The same argument can be made for the basket. Indeed, the basket is required for SAGA-dependent recruitment of chromatin, including circles, to NPCs (***Dieppois et al., 2006***; ***Shcheprova et al., 2008***; ***Texari et al., 2013***). Therefore, circle-bound NPCs must have lost their basket only during or after circle attachment. Remarkably, the allele Nup60-K467R, which is not displaced from NPCs upon circle binding, stabilizes Nup116 on circle-bound NPCs, indicating that Nup116 is most probably released

from NPCs after Nup60 acetylation, that is, upon circle docking (see below). Thus, our data suggest that NPCs become extensively remodeled upon DNA circle attachment rather than being incomplete in the first place.

Interestingly, all Nups lacking from old cells and circle-bound NPCs have been shown to turnover at NPCs by microscopy (*Denoth-Lippuner et al., 2014*) or mass-spectrometry methods (*Hakhverdyan et al., 2021*): The basket components Nup2 and Nup60, and the cytoplasmic complex component Gle1 are highly mobile at NPCs. Nup116, Nup42, Gle2, Mlp1, and Mlp2 turnover a bit slower but are also mobile. In contrast, all core Nups which are stable at the NPC cap fail to show turnover by mass-spectrometry. The same applies for Nup82 and Nup159, the NPC components on the cytoplasmic side that are not displaced from circle-bound NPCs (*Hakhverdyan et al., 2021*). We noticed only two exceptions. Nup1 and Nup53 are mobile but remain associated with circle-bound NPCs (*Hakhverdyan et al., 2021*) indicating that circles do not displace all dynamic Nups, but only specific ones. Together, these data show that circle-bound NPCs are not affected in their core. Only the dynamics of specific mobile components seem to be changed.

More importantly, the observation that removal of the basket and cytoplasmic complex involves the acetylation of Nup60, a SAGA target, indicates that this remodeling is not incidental but regulated. In young cells, SAGA promotes the docking of many genes to NPCs upon their activation (*Cabal et al., 2006*; *Huisinga and Pugh, 2004*; *Kremer and Gross, 2009*; *Luthra et al., 2007*) and Nup60 acetylation controls the expression of diverse genes (*Kumar et al., 2018*). The residence of the basket at NPCs is also dynamic in young cells (*Akey et al., 2022*; *Denoth-Lippuner et al., 2014*; *Hakhverdyan et al., 2021*). Thus, rather than a damage, the acetylation-dependent removal of the basket in circle-bound NPCs seems to reflect a regulatory event, also frequent in young cells, before circles are formed.

The idea that the removal of the basket and cytoplasmic components upon circle-binding is a regulatory event is further supported by the fact the interaction of these NPCs with karyopherins is affected for only few of them, all specifically involved in mRNA export. Thus, a parsimonious but exciting hypothesis is that circle attachment promotes regulatory events that specialize the affected NPCs for protein transport, inhibiting mRNA export. Thus, we suggest that ERC attachment leads to the accumulation of too many specialized NPCs. Future studies will address how this imbalance affects cellular viability. However, our results suggest specific avenues for research. Specifically, the increased import of protein and possibly a defective export of mRNAs (although the properly assembled NPCs remaining might be sufficient for this function) may increase crowding in the nucleus and explain the misregulation of ribosome and nucleosome assembly observe in old cells (*Janssens et al., 2015*; *Morlot et al., 2019*).

## Functional diversity of NPCs

The idea that NPCs might exist in different flavors is not novel (*Casolari et al., 2004*). In budding yeast, the NPCs adjacent to the nucleolus have no basket (*Galy et al., 2004*; *Zhao et al., 2004*) and all yeast NPCs lose their basket in response to heat stress (*Carmody et al., 2010*), documenting that basket removal is indeed a recurrent mode of NPC regulation. The remodeling that we report here is even more profound, as it also involves the cytoplasmic complex. We suggest that our observation provides insights into the compositional and functional plasticity of NPCs and establish that their acetylation is an important mode of their regulation.

## And beyond yeast?

There is currently no evidence for DNA circles contributing to ageing in other organisms but there is strong evidence that chromatin interaction with the nuclear periphery is affected by age in many cell types. Most remarkably, progerin, a progeriatric isoform of Lamin A and causing the Hutchinson-Gilford progeria syndrome in humans, displaces Nup153 and TPR (the mammalian homologs of Nup60 and Mlp1/2) from the nuclear periphery (*Balmus et al., 2018*; *Cobb et al., 2016*; *Kelley et al., 2011*; *Larrieu et al., 2018*). Thus, the effects of ageing on NPCs and the role of NPCs in ageing might be strikingly similar between yeast and mammals.

The notion that age-dependent remodeling of NPC involves physiologically relevant regulatory steps, and not damage, might appear surprising. We would like to propose that it is however one of the modalities predicted by the antagonistic pleiotropy theory, which may explain the emergence of

ageing during evolution (*Kirkwood and Rose, 1991*). This theory suggests that ageing emerges upon fixation of traits and processes that have selective advantage early in life, despite being deleterious later-on. This theory underlays the wide spread idea that sparing on quality control and damage-repair liberates resources for the generation of progeny early in life, at the expense of longevity (*Ackermann et al., 2007*; *Austad and Hoffman, 2018*; *Williams, 1957*). The case of NPCs and their role in SAGA-dependent gene regulation suggest that another possible modality of antagonistic pleiotropy concerns tradeoff effects associated with adaptability. The data available suggest that the ability to restructure and specialize NPCs plays an important role in how cells adapt to a variety of stresses and environmental changes. Our data suggest that this essential function happens at the cost of longevity.

## Materials and methods

### Strains and plasmid

All the yeast strains and plasmids used in this study are listed in *Supplementary file 1* and are isogenic to S288C. GFP-tag and knock-out strains were generated using classical genetic approaches (*Janke et al., 2004*). All cultures were grown using standard conditions, in synthetic drop-out medium (SD-medium; ForMedium, Norfolk, UK) or indicated otherwise, at 30 °C.

The non-chromosomal DNA circle was obtained from the Megee lab (*Megee and Koshland, 1999*) and contains an array of 256 TetO repeats, the centromere is flanked by target site for the R-recombinase; the β-estradiol-inducible expression recombinase (from the genome) drives the excision of the centromere and converts the minichromosome into a non-chromosomal DNA circle (*Baldi et al., 2017*; *Denoth-Lippuner et al., 2014*; *Shcheprova et al., 2008*). TetR-NLS-mCherry is genetically expressed and labels the TetO repeats; the NLS (nuclear localization signal) ensures its accumulation in the nucleus.

The nup60-K467R strain was generated with a one-step CRISPR-Cas9 method, based on *Laughery et al., 2015*. The gRNA was designed with an online tool to identify an optimal guide RNA (gRNA) target site in Nup60 (http://wyrickbioinfo2.smb.wsu.edu/crispr.html). The donor DNA with the Nup60K467R mutation and the gRNA-encoded DNA were ordered as single stranded oligo's (Microsynth AG, Balgach, Switzerland) and annealed. The gRNA-encoded DNA was recombined into the Cas9 expression vector pML104_Cas9_HygR in a one-step approach, to induce the Nup60K467R mutation simultaneously: SwaI-linearized vector, the gRNA-encoded double stranded oligos and the donor DNA were transformed all at once into a wild type yeast strain and plated on hygromycin selection medium. The expression vector with gRNA (pML104_Cas9_HygR_Nup60) was rescued from cell material, propagated in bacteria and confirmed by digest analysis and sequencing. Genomic DNA was extracted from single yeast clones and the presence of the point mutation was confirmed by sequencing. Positive clones were propagated on YPD to get rid of the expression vector.

### Microscopy

For fluorescent microscopy, yeast cells were precultured for minimally 24 hr in synthetic drop-out medium. One ml of cells from exponential growing cultures with OD <1 was concentrated by centrifugation at 1.000xG, resuspended in ~5 µl of low fluorescent SD-medium, spotted on a round coverslip and immobilized with a SD/agar patch. The cells were imaged in z-stacks of 6 slices with 0.5 µm spacing, with a 100×/1.4 NA objective on a DeltaVision microscope (Applied Precision) equipped with a CCD HQ2 camera (Roper), 250 W Xenon lamps, Softworx software (Applied Precision) and a temperature chamber set to 30 °C.

To accumulated DNA circles in the nuclei of ageing mother cells, yeast cells were pre-cultured for 24 hr in SD–URA at 30 °C and then shifted to SD-LEU medium supplemented with 1 µM β-Estradiol (Sigma-Aldrich, St. Louis, MO), incubated for 16–18 hr at room temperature. The β-Estradiol induced expression of the recombinase and the excision of the centromere, see:(*Denoth-Lippuner et al., 2014*). To visualize DNA clusters, we use specifically 1 × 1 binning and made short time-lapse movies of 15 min, intervals for 5 min.

### Ageing microfluidic platform

Nucleoporin colocalization and the tandem fluorescent protein timer analysis during ageing were investigated using the high-throughput yeast ageing analysis (HYAA) microfluidics dissection platform

(*Jo et al., 2015*). The PDMS (polydimethylsiloxane) microchannel is made by soft-lithography and bonded on the 30 mm micro-well cover glass in the 55 mm glass bottom dish (Cellvis, CA, USA). For the lifespan analyses, a chip with a new cell trapping design was used (*Figure 6A and B*), to ensure excellent retention of old cells (see below).

To start the experiment, yeast cells were pre-cultured for 24 hr in SD-full supplemented with 0.1% Albumin Bovine Serum (protease free BSA; Acros Organics, Geel, Belgium). Young cells from an exponentially growing culture were captured in the traps of the microfluidic chip; the chip was continuously flushed with fresh medium at a constant flow of 10 µl/min, using a Harvard PHD Ultra syringe pump (Harvard Apparatus, Holiston, MA, USA) with two or four 60 ml syringes, with inner diameter 26.7 mm (Becton Dickinson, Franklin Lakes, NJ, USA). Bright field images were recorded every 15 min. throughout the duration of the entire experiment. To measure the nucleoporin colocalization, fluorescent images only after 2 hr, 12 hr, 26 hr or/and 50 hr. For imaging we used an epi-fluorescent microscope (TiE, Nikon Instruments, Tokyo, Japan) controlled by Micro-Manager 1.4.23 software (*Edelstein et al., 2014*), with a Plan Apo 60 × 1.4 NA objective. For fluorescence illumination of the GFP and mCherry labeled proteins, a Lumencor Spectra-X LED Light Engine was used. Stacks of 7 slices with 0.3 µm spacing were recorded during. The age of the cell was defined by the number of daughter cells that emerged during the budding cycles.

For the nucleoporin correlation analysis, a cell of interest was manually selected if it stays in the focal plane in the bright field channel. Its age was determined and a segmented line was drawn through the nuclear envelope in an image in the focal plane, using Fiji/ImageJ 1.51 n (*Schindelin et al., 2012*), and the intensity profiles were recorded for both fluorescence channels. The Pearson correlation between the intensity profiles was calculated and plotted in R (*R Development Core Team, 2021*).

For the tandem fluorescent protein timer analysis, late mitotic cells were selected after 2 hr and 26 hr incubation in the chip. Its age was determined and a segmented line was drawn through the nuclear envelope in an image in the focal plane, as described above. The average background corrected intensity for GFP and mCherry was calculated and plotted in R.

To obtain reliable lifespan curves, the majority of the cells should be retained until their death, to prevent biasing the data. Although different microfluidic dissection platforms have been developed, it is still a challenge to reach high enough retention efficiency in the microfluidics chip for life span analysis. Here we used an improved design of yeast cell traps, having small 'claws' at both sides, preventing the escape of bigger cells at higher age (*Figure 8A and B*). This allowed us to retain >95% of the cells during their full lifetime. Only bright field images were recorded every 15 min. throughout the entire experiment of 70–80 hr. All cells in a field of view were analyzed, the replicative lifespan was determined for each single cell by counting the budding cycles before cell death.

## FLIP experiments (fluorescence loss in photobleaching)

For FLIP experiments, centromere excision and plasmid accumulation was achieved as previously described. Cells were immobilized on a 2% agar pad containing SD medium. Images were acquired on Confocal Zeiss LSM 880 Airyscan microscope, Alpha PA 63 x/1.40 Oil objective, using ZEN software with 2% 488 nm laser power, timeframes of 500ms, on a single Z plane. A nuclear ROI of choice was bleached for 10 iterations over 13 s with 100% 488 nm laser power (~500 mV). Emission signals were collected on 32-channel GaAsP-PMT (gallium arsenide phosphide photomultiplier tube) Airy detector. Intensity of the GFP signal was quantified in ImageJ/FIJI software, at the Nuclear Envelope ROI next to the plasmid aggregate, a second ROI on the nuclear envelope (NE) at equal distance from the bleach point, as well as in an adjacent cell to control for overall sample bleaching. Signals are plotted as relative to the pre-bleaching frame. At least 30 cells were analyzed per strain.

## Acknowledgements

We thank Joachim Hehl, Tobias Schwartz and the Scientific Center for Optical and Electron Microscopy (ScopeM) at ETH Zürich for microscopy support; Rodrigo Merayo Martínez, Irina Remple and Liesbeth Veenhoff for their help with running the microfluidic chips. We thank Karsten Weis and members of the Barral lab for helpful discussions and comments on the manuscript. We acknowledge financial support by Swiss National Science Foundation (grant 31,003 A-105904 to YB) and the European Union's Horizon 2020 Research and Innovation program under grant agreement N° 899417/Circular Vision project.

## Additional information

### Funding

| Funder | Grant reference number | Author |
|---|---|---|
| Schweizerischer Nationalfonds zur Förderung der Wissenschaftlichen Forschung | 31003A-105904 | Yves Barral |
| H2020 European Research Council | 899417 | Yves Barral |

The funders had no role in study design, data collection and interpretation, or the decision to submit the work for publication.

### Author contributions

Anne C Meinema, Investigation, Methodology, Visualization, Writing – original draft; Anna Marzelliusardottir, Mihailo Mirkovic, Investigation, Writing - review and editing; Théo Aspert, Sung Sik Lee, Gilles Charvin, Methodology; Yves Barral, Conceptualization, Funding acquisition, Supervision, Writing - review and editing

### Author ORCIDs

Anne C Meinema (iD) http://orcid.org/0000-0002-0002-3486
Anna Marzelliusardottir (iD) http://orcid.org/0000-0003-0887-4741
Théo Aspert (iD) http://orcid.org/0000-0003-2957-0683
Sung Sik Lee (iD) http://orcid.org/0000-0001-9267-232X
Gilles Charvin (iD) http://orcid.org/0000-0002-6852-6952
Yves Barral (iD) http://orcid.org/0000-0002-0989-3373

### Decision letter and Author response

Decision letter https://doi.org/10.7554/eLife.71196.sa1
Author response https://doi.org/10.7554/eLife.71196.sa2

## Additional files

### Supplementary files

• Supplementary file 1. Table containing all the yeast strains and plasmids used in this paper.
• Transparent reporting form

### Data availability

All data generated or analysed during this study are included in the manuscript and supporting files.

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
