## [Editor Report]

This interesting study examines a potential relationship between the tethering of extrachromosomal DNA (ecDNA) to the nuclear pore complex (NPC) and its role in aging; a model is proposed whereby the nuclear basket is displaced by ecDNA anchoring, which leads to a broader remodeling of the NPC that is distinct from NPC damage. This idea is conceptually novel and will represent an important advance, although some more support for the conclusions is still needed.

---

## [Decision Letter]

**Decision letter after peer review:**

Thank you for submitting your article "Specialization of chromatin-bound nuclear pore complexes promotes yeast aging" for consideration by *eLife*. Your article has been reviewed by 3 peer reviewers, one of whom is a member of our Board of Reviewing Editors, and the evaluation has been overseen by Matt Kaeberlein as the Senior Editor. The following individual involved in review of your submission has agreed to reveal their identity: Hermann Broder Schmidt (Reviewer #2).

Essential revisions:

1) All three reviewers wanted some further clarification or better characterization on remodeled NPCs, including some of its components and transporters.

2) All three reviewers pointed out various issues related to the involvement of SAGA activity or SAGA related factors in NPC remodeling. Please address these concerns in your revision.

3) All three reviewers specified some wording and reference problems. Please consider them when revising your manuscript.

*Reviewer #1 (Recommendations for the authors):*

Here are a few major concerns that if addressed would improve the overall story of the manuscript:

1. "chromatin-bound nuclear pore complexes" in the title is not accurate. DNA circles, including rDNA circles found in aged cells, are not part of the chromatin and may or may not even be packaged with nucleosomes. The authors should consider revising it.

2. Figure 4 is probably the least convincing data in this manuscript and analysis should be repeated for an additional well-defined marker of dysfunctional NPCs, such as those for storage of improperly assembled NPCs compartment (SINC). SINC is known to segregate with the mother cells like rDNA circles. So whether SINC and DNA circles (NPC caps) colocalize in aged cells should be examined.

3. Representative images for Figure 4B should be included.

4. The TREX2 complex (Thp1-Sac3-Sus1-CDC31) is a well-known SAGA-associated factor involved in mRNA export. Given the data showing that DNA circle bound NPCs are specifically depleted of factors involved in mRNA export and SAGA is likely involved in NPC remodeling, the association of TREX2 component with remodeled NPCs should be assessed like the transport factors examined in Figure 7.

*Reviewer #2 (Recommendations for the authors):*

Although I enjoyed the science, I unfortunately find that the paper is not written very clearly. This is especially the case in the introduction, which poses many more big picture questions than the paper actually addresses. In my opinion, focusing the introduction on a single question/hypothesis would greatly contribute to overall clarity. The authors also make generous use of adverbs to start their sentences, which often is more confusing than helpful. This includes the beginning of subsections with the word 'thus' (page 8, line 20; page 9, line 1) and the five-sentence ping-pong sequence beginning with 'accordingly', 'on the opposite', 'accordingly', 'still', 'however' (page2, lines 5-14). I strongly encourage the authors to fine-tune their writing to improve the conceptual clarity of their paper. Specific arguments that I find hard to follow are:

- Page 7, lines 28-35. Do I understand correctly that the authors want to distinguish between the models that: (a) the SAGA complex is structurally required for tethering ERCs and (b) acetyl-transferase activity is necessary? If so, can a (point) mutation of GCN5 or small molecule drug that inactivate the enzyme be used as an alternative to the knock-out?

- Page 9, lines 17-23. The authors seem to suggest that Kap60 and Kap123 get stuck in the pores, thus explaining the reduction of ribosome and nucleosome assembly in aged cells. What's the evidence for this? I think it's important to distinguish between the enhanced shuttling and getting stuck models. Perhaps photoswitchable fluorescent proteins would offer the necessary spatiotemporal control for in vivo transport assays.

*Reviewer #3 (Recommendations for the authors):*

1) There is little direct data that tethering of ecDNA drives NPC remodeling but throughout the text it is consistently repeated that ecDNA tethering to NPCs drives or "displaces" the nuclear basket, however, the data demonstrating this is correlative. Thus, at a minimum, this language should be rephrased throughout and toned down to better reflect the data or additional experiments would be required to more clearly demonstrate that ecDNA tethering directly displaces the nuclear basket (and cytosolic nups).

2) The concept of "NPC specialization" is interesting but the suggestion is to be a bit more circumspect to alternative explanations for the lack of basket/cytosolic filament components in the NPC cap. Several reasons: first, as individual NPCs cannot be resolved, it cannot be differentiated whether all the NPCs in the cap are compositionally identical. i.e. the loss of certain nups could represent a reduction in the stoichiometry of some nups in some NPCs and need not represent a new class of specialized NPC (see also comment 4). Second, it cannot be differentiated (based on the current data) whether the NPCs in the cap are a product of nup loss or defective NPC assembly. With the latter in mind, in a model in which the cap has defectively formed NPCs, one could imagine a scenario in which they aren't formed correctly because there is simply less of a given nup produced in the mother cell (or perhaps some are targeted by recently discovered autophagy pathways that target Nup159, e.g. Tomioka et al., 2020; Lee et al., 2020). Evaluating the total levels of the Nup fusions, at least at the level of total fluorescence, could help rule out this possibility.

3) SAGA is thought to be bound to many NPCs irrespective of ecDNA where it presumably acetylates Nup60 as suggested by the authors. Can the authors clarify whether it is SAGA, or the tethering of ecDNA to SAGA, that triggers the putative remodeling? As I can't think of a straightforward experiment to test this, I suggest at least discussing caveats associated with the model along this point.

4) There are several results that may be inconsistent with the literature/our understanding of NPC structure that need to be considered and/or explained: a) That Nup159 and Nup82 remain at NPCs while Nup116 is missing (Figure 1) is incongruent with work suggesting that at least Nup159 is missing from nup116Δ NPCs (see Allegretti et al., 2020). b) The localization of NTRs particularly Kap60 is a bit confusing as in the absence of Nup60 it localizes in the nucleus not the nuclear periphery (Denning et al., 2001) as Nup2 is also mislocalized in this strain. Thus it seems counterintuitive that Kap60 would be found in the NPC cap as suggested in Figure 7.

[Editors' note: further revisions were suggested prior to acceptance, as described below.]

Thank you for resubmitting your work entitled "DNA circles promote yeast ageing in part through stimulating the reorganization of nuclear pore complexes" for further consideration by *eLife*. Your revised article has been evaluated by Jessica Tyler (Senior Editor) and a Reviewing Editor. The following individual involved in the review of your submission has agreed to reveal their identity: Hermann Broder Schmidt (Reviewer #2). The reviewers have discussed their reviews with one another, and the Reviewing Editor has drafted this to help you prepare a revised submission.

The manuscript has been improved but there are some remaining issues that need to be addressed, as outlined below:

Essential revisions:

Reviewer 3 asks for some further clarification on SINC (see the comments below). Please address or respond to this remaining concern.

*Reviewer #1 (Recommendations for the authors):*

The authors have addressed all the concerns raised by reviewers in this revised manuscript. I recommend accepting this manuscript for publication in *eLife*.

*Reviewer #2 (Recommendations for the authors):*

The thorough revisions strengthen the manuscript further, and I can't wait to see this nice work published in *eLife*.

*Reviewer #3 (Recommendations for the authors):*

The authors were able to address the majority of my concerns in the revised manuscript. Some clarification is needed about the SINC, however. The SINC was first described in Webster et al., Cell, 2014 as an accumulation of misassembled NPCs that arises upon deletion of VPS4 or SNF7 and is retained in mother cells. Thus, neither Vps4 nor Snf7 localizes to the SINC as suggested by the authors (but Chm7 does, see Webster et al., 2016). Further, the idea that the SINC actively recognizes defective NPCs or that anything is "targeted by the SINC" is probably inaccurate and should be reworded to "targeted by ESCRTs". For example, there is evidence that ESCRTs like Chm7 (but also likely Snf7 and Vps4) localize to the nuclear envelope when NPC assembly is perturbed (Webster et al., 2016). Thus, these and other data have implicated ESCRTs in a form of NPC quality control. To be clear, nothing needs to change with regards to the data itself but I suggest that the authors are a bit more circumspect when labeling the Snf7/Vps4 focal nuclear envelope localization as SINCs.

---

## [Author Response]

Essential revisions:1) All three reviewers wanted some further clarification or better characterization on remodeled NPCs, including some of its components and transporters.2) All three reviewers pointed out various issues related to the involvement of SAGA activity or SAGA related factors in NPC remodeling. Please address these concerns in your revision.3) All three reviewers specified some wording and reference problems. Please consider them when revising your manuscript.

In the original submission we wrote that the composition of the NPC alters upon the attachment of extrachromosomal DNA circles in old yeast cells. Specifically, the interaction with DNA circles displaces the peripheral subunits from the core of the NPC, leaving the pores without basket and cytoplasmic complexes. We proposed that displacement was not the result of damage, but rather a regulated remodeling of the NPCs. These modifications affected the interaction with mRNA export factors specifically, without changing the residence of import factors. Mutations preventing the remodeling of the NPCs extended the lifespan of the cells. We concluded that DNA circle accumulation during aging in mother cell drive aging, at least in part, via NPC modulation.

Although the overall positive feedback, some reviewers raised concerns about the conclusion we had drawn from it. We have addressed all these issues. In particular we have done the following:

1. We have additionally studied the dynamics of transport factors in the circle-bound NPCs, which are accumulated in the caps at the DNA cluster. Reviewer 2 pointed rightfully out that the displacement of certain Nups might affect the integrity of the NPC’s permeability barrier strongly, leading to a potential collapsing of the RanGTP gradient and preventing transport across the central channel of the circle-associated NPCs. Without a RanGTP gradient, transport factors will not be dissociated from the FG-Nups and ultimately getting stuck in the pores. This would lead to an accumulation of transport factors in these pores. Although we did not observe this for the majority of the transport factors in our studies, two import factors accumulated in circle-bound NPCs (Kap60 and Kap123). To investigate directly whether transport factors got immobilized in these NPC, we measured the dynamicity of these two NPC accumulated transport factors by FLIP. We observed no significant difference for the dynamics of both transport factor in pores localized in the cap at the DNA clusters compared to the NPCs in the rest of the nuclear envelop (new figure 6A-B). This data shows that the transport factor exchange in circle-bound NPCs is comparable to the ones without the association of DNA circles, located in the rest of the nuclear envelop. Thus, we assume that, although the displacement of several important Nups, the RanGTP gradient is not affected in these pores.

2. We have furthermore expanded our studies on whether the circle-bound NPCs are defective and recognized by the mechanism to remove damaged or misassembled NPCs or are rather remodeled via posttranslational modifications. As indicated by reviewer 3, this is indeed a challenging idea, and we do not want to stretch our claims here. We have adjusted the manuscript to explain better that we assume that the remodeling of the NPC indeed might have subsequent damaging consequences for the NPCs and the physiology of the cell. The displacement of the mRNA export factors from these NPCs are indeed indicative for a malfunction of these pores and might have drastic impact on protein synthesis rates. However, what we propose is that the displacement of the peripheral subunits itself is a regulated modulation of the NPC, important for its function to retain DNA circles in the mother cell. We have adapted the text to clarify our conclusion better and we added additional experiments to test the idea that circle-bound NPC are indeed not damaged. These new data indeed support our conclusion

a. First, we studied the localization of additional components of the storage of improper assembled NPCs compartment (SINC) in respect to DNA circles in old mother cells. The SINC represents a quality control system that recruits the ESCRT III machinery to detect and remove defective NPCs in the nuclear envelope. These SINCs were shown to accumulate in mother cells. However, when we studied the location of the SINC components in old cells on the microfluidics chip, we did not see the SINC proteins colocalizing with DNA circle clusters (Figure 4A-C). Although damaged NPCs accumulate in old cells, our data showed no enrichment for these damaged NPCs at the DNA circle clusters. We interpreted this data that DNA circle-loaded NPCs thus are not recognized by the ESCRT III machinery as being defective.

b. We next investigated the possibility that circle-bound NPCs of old cells are recognized and targeted by the SINC. Thus, we used the fact that ERCs bind the protein Net1 to ask whether the SINC accumulates to the vicinity of ERC-bound NPCs. However, this was not the case either. These data are now shown in figure 4D-E.

c. The new data showing that the dynamics of Kap60 and Kap123 is not affected in circle-bound NPCS (see above) support as well the notion that these NPCs are dramatically defective.

d. We have added a new experiment to confirm that the NPCs of old cells are not leaky, as already observed by others (see Morlot et al., 2019; Rempel et al., 2019).

Together, the additional data did not indicate that DNA-bound NPCs in old cells show a sign of any immediate defect. This supports our initial idea that these NPCs are specialized for DNA circle retention in the mother cell. However, we acknowledge that the progressive accumulation of many of these modified NPCs can be considered to be aging-induced defect in the cell.

3. Finally, we discussed and studied in more detail the cause and consequence relation between DNA circle attachment and basket displacement. This concern was raised by reviewer 3, asking whether altered (i.e. defective) NPCs are more present in the old cells and that they could attract DNA circles, rather than DNA circles displace the peripheral structures from the NPC. We now discuss this point more in depth and make several points further supporting our initial conclusions. First, we have noticed in an earlier study (Denoth-Lippuner et al., e*Life*, 2014) that acetylation of Nup60 is required for DNA circle binding to the NPC. This speaks for a specific regulated posttranslational modification of Nup60 for DNA circle binding, induced by the circle association with the pore, considering that the circle is bound with SAGA’s acetyltransferase Gcn5. A random aging-induced alterations is unlikely to bind DNA circles in such a regulated fashion. Second, as we previously showed, DNA circles no-longer colocalize with NPCs in *mlp1∆ mlp2∆* double mutant cells and DNA circle are no-longer confined to these mutant mother cells (Shcheprova Z. et all, Nature 454:728–734). This implies that NPCs without the basket cannot attach to DNA circles anymore. How this mechanistically works is up for further investigation, but it at least indicates that the basket is involved in the interaction of circles at the NPCs. Thus, our observation that the basket is displaced from circle-bound NPCs indicate that this displacement is subsequent to circle-binding. Likewise, the Nups Nup82 and Nup159 have been recently shown to require Nup116 for their recruitment to the NPC. The fact that these Nups are not displaced from circle-bound NPCs but Nup116 is argue for Nup116 being displaced upon circle binding rather than being absent in the first place. Accordingly, we show that preventing Nup60 acetylation, which our data identify as a target of SAGA upon circle binding, restores Nup116 localization. Thus, Nup116 displacement seems to be a consequence of Nup60 acetylation upon circle anchorage to NPCs.

Thus, the most parsimonious hypothesis for explaining these different observations is that DNA circle anchoring to the NPC core drives the displacement of peripheral subunits, starting by the nuclear basket.

We thank the reviewers for their valuable comments on our manuscript. We believe that we have covered all the concerns raised and hope for a smooth publication in *eLife*.

Reviewer #1 (Recommendations for the authors):Here are a few major concerns that if addressed would improve the overall story of the manuscript:1. "chromatin-bound nuclear pore complexes" in the title is not accurate. DNA circles, including rDNA circles found in aged cells, are not part of the chromatin and may or may not even be packaged with nucleosomes. The authors should consider revising it.

We have changed our title to “DNA circles promote yeast aging in part through stimulating the reorganization of nuclear pore complexes”, which indeed summarizes our observations more conservatively. However, we would like to point out that several observations indicate that DNA circles are chromatinized in vivo.

1. Studies of the Thoma lab has mapped with great precision the position of nucleosomes on replicating circles in vivo, back in the 90’s and 2000’s (Tanaka et al., J. Mol. Biol., 1996; Suter et al., NAR, 2000; Suter and Thoma, JMB, 2002). For a given gene (URA3 in this case), nucleosome positioning did not vary substantially between the circle and chromosomal context. The plasmids used did not contain any centromere, putting them in the same category as bona-fide DNA circles popping out from the chromosome.

2. Consistent with this notion, the recruitment of SAGA onto DNA circles and its role in linking them to NPCs indicates that these circles are packed into nucleosomes, since SAGA relies largely on nucleosome-binding to indirectly bind DNA.

3. In line with this, we showed that mutating S10 on H3 modulates in an Hst2-dependent manner the compaction state of DNA circles. This effect relies on H3 deacetylation (Wilkins et al., 2014; Kruitwagen et al., 2018) and hence on the presence of nucleosomes.

2. Figure 4 is probably the least convincing data in this manuscript and analysis should be repeated for an additional well-defined marker of dysfunctional NPCs, such as those for storage of improperly assembled NPCs compartment (SINC). SINC is known to segregate with the mother cells like rDNA circles. So whether SINC and DNA circles (NPC caps) colocalize in aged cells should be examined.

We have followed the reviewer’s recommendation by adding two more markers of the SINC, Vps4 and Snf7, and by doing these measurements both for cells accumulating our reporter circle and for old cells aged and imaged in our microfluidic chip. In these later cells, the accumulated ERCs are visualized using the Net1 protein fused to mCherry. Net1 is recruited to the RFB site in the rDNA unit and labels both the chromosomal rDNA locus and the extrachromosomal rDNA circles (see Neurohr et al., Genes Dev., 2018; Morlot et al., 2019). In old cells, the accumulation of ERCs is visible as a nearly 10-fold growth of the Net1-labeled area in the nucleus (new Figure 4). In either case, none of the two SINC markers decorated circle- or rDNA-associated NPCs more than bulk ones.

3. Representative images for Figure 4B should be included.

This is now done (new figure 4F). As indicated by the quantifications, there is not much to be seen.

4. The TREX2 complex (Thp1-Sac3-Sus1-CDC31) is a well-known SAGA-associated factor involved in mRNA export. Given the data showing that DNA circle bound NPCs are specifically depleted of factors involved in mRNA export and SAGA is likely involved in NPC remodeling, the association of TREX2 component with remodeled NPCs should be assessed like the transport factors examined in Figure 7.

We have now added the data for Sac3, which in average is indeed excluded from the NPC-cap covering the cluster of circles, although we sporadically find it enriched within the clusters. Furthermore, the cluster tends to be fragmented in the cells expressing Sac3-GFP, suggesting that tagging might affect its function. Further studies will be required for clarifying that case more precisely.

Reviewer #2 (Recommendations for the authors):Although I enjoyed the science, I unfortunately find that the paper is not written very clearly. This is especially the case in the introduction, which poses many more big picture questions than the paper actually addresses. In my opinion, focusing the introduction on a single question/hypothesis would greatly contribute to overall clarity. The authors also make generous use of adverbs to start their sentences, which often is more confusing than helpful. This includes the beginning of subsections with the word 'thus' (page 8, line 20; page 9, line 1) and the five-sentence ping-pong sequence beginning with 'accordingly', 'on the opposite', 'accordingly', 'still', 'however' (page2, lines 5-14). I strongly encourage the authors to fine-tune their writing to improve the conceptual clarity of their paper. Specific arguments that I find hard to follow are:- Page 7, lines 28-35. Do I understand correctly that the authors want to distinguish between the models that: (a) the SAGA complex is structurally required for tethering ERCs and (b) acetyl-transferase activity is necessary? If so, can a (point) mutation of GCN5 or small molecule drug that inactivate the enzyme be used as an alternative to the knock-out?

Actually, we have already show in a previous study that both the interaction of SAGA with NPCs and its acetyltransferase activity (using a Gcn5 catalytically inactive point-mutant) are required for tethering DNA circles to NPCs (Denoth-Lippuner et al., 2014). The question that we want to ask here is whether the acetyltransferase activity of SAGA contributes to the absence of the basket on these NPCs. This is a different question.

Unfortunately, the role of SAGA in Nup60 acetylation cannot be addressed using a point mutation in Gcn5 or inhibitors, since these perturbations prevent the attachment of the circles to NPCs in the first place. For this reason, we have relied on mutating the target site of the enzyme in Nup60, which controls the recruitment of all other basket components to NPCs. We have rephrased the text to improve clarity: The text is now focused on the role of acetylation in Nup60 localization, specifically. To avoid confusion, we no-longer mention the role of SAGA on circle attachment, which is already established.

- Page 9, lines 17-23. The authors seem to suggest that Kap60 and Kap123 get stuck in the pores, thus explaining the reduction of ribosome and nucleosome assembly in aged cells. What's the evidence for this? I think it's important to distinguish between the enhanced shuttling and getting stuck models. Perhaps photoswitchable fluorescent proteins would offer the necessary spatiotemporal control for in vivo transport assays.

Thank you for raising this point. We have now addressed it, using fluorescence loss in photobleaching (FLIP). In short, in strains where either Kap60 or Kap123 was tagged with GFP we repeatedly photobleached a small area in the center of the nucleus and monitored the loss of fluorescence at the nuclear periphery. This experiment was carried out in cells that had accumulated the reporter plasmid and formed a circle cluster covered by an NPC cap. We monitored and compared the fluorescence decay in both the cap and elsewhere in the envelope. The fluorescence decay shows undistinguishable kinetics between the two areas. Thus, neither of the two importins is particularly stuck in the circle-bound NPCs compared to bulk NPCs.

Reviewer #3 (Recommendations for the authors):1) There is little direct data that tethering of ecDNA drives NPC remodeling but throughout the text it is consistently repeated that ecDNA tethering to NPCs drives or "displaces" the nuclear basket, however, the data demonstrating this is correlative. Thus, at a minimum, this language should be rephrased throughout and toned down to better reflect the data or additional experiments would be required to more clearly demonstrate that ecDNA tethering directly displaces the nuclear basket (and cytosolic nups).

Our original manuscript combined results and discussion, which might have caused some confusion. We have now separated the two from each other, and discussed in more depth the different options for why and how circle-bound NPCs lack a basket.

Concerning the substance of this discussion, the hypothesis that most parsimoniously accounts for all currently available observations remains that circle attachment leads to basket displacement rather than circles attaching to NPCs that are already basketless. We explain now in the Discussion section why we favor this interpretation. Here are our arguments:

The initial attachment of DNA circles to NPCs, and of any chromatin, requires the presence of the basket. This is well documented for the chromosomal loci moving to the nuclear periphery upon their induction (See Dieppois et al., 2006; Texari et al., 2013). We have reported the same for DNA circles, which also attach to basket-carrying NPCs (Shcheprova et al., 2008). Accordingly, cells lacking both Mlp1 and Mlp2 fail to retain DNA circles in the mother cell. Thus, the removal of the basket must be a secondary event taking place after attachment.

In favor of this notion, we show that removal of Nup60 from circle-bound NPCs requires its acetylation, a modification carried out by the SAGA complex. The attachment of the circles to NPCs is mediated by the SAGA complex both as a structural linker (through its NPC-binding subunits Sgf73 and Sus1) and as a modifying enzyme (through its acetyltransferase activity, mediated by the catalytic site of its catalytic subunit, Gcn5; Denoth-Lippuner et al., *eLife*, 2014). Accordingly, the SAGA complex binds both the DNA circles and the NPCs, and accumulate in the circle cluster (Denoth-Lippuner et al., 2014). Thus, the simplest interpretation is that recruitment of the DNA circles to the nucleoplasmic side of the NPCs by the basket recruits SAGA, which subsequently triggers basket removal by modifying Nup60 and probably other basket components.

Lending further support to this hypothesis, we show that the accumulation of basketless NPCs as cells age requires both circle formation and SAGA function, particularly its NPC-targeting subunit, Sgf73. This is fully consistent with the idea that basket removal depends on SAGA function upon circle-binding.

Finally, we would like to point out that our data correlate relatively well with the recently published data of the Rout lab, identifying the nucleoporins that are the most mobile at properly assembled NPCs. Indeed, while we do not find all mobile Nups being displaced from cap NPCs, all the Nups that we find as being displaced from the cap are mobile Nups. As we will discuss below, the fact that a Nup such as Nup116 is absent from circle-bound NPCs, whereas Nup82 and Nup159, which rely on Nup116 for insertion in the pore complex, are present suggests that Nup116 must have been at the NPCs sometime before circle docking. Thus, these Nups are indeed more likely to be removed upon circle docking rather than lacking in first place.

Likewise, since the absence of the cytoplasmic complex, including Nup116, on circle-bound NPCs relies on the modification of Nup60 by SAGA, it is likely that both Nup60 and the cytoplasmic complex must have been at these NPCs before docking of the circles.

However, we fully acknowledge the fact that we are not in position yet to directly capture and image the events of circle attachment to NPCs. We cannot directly show at this stage that circle recruitment displaces the basket. As we write, this only a hypothesis, but the hypothesis that most parsimoniously account for the available data.

2) The concept of "NPC specialization" is interesting but the suggestion is to be a bit more circumspect to alternative explanations for the lack of basket/cytosolic filament components in the NPC cap. Several reasons: first, as individual NPCs cannot be resolved, it cannot be differentiated whether all the NPCs in the cap are compositionally identical. i.e. the loss of certain nups could represent a reduction in the stoichiometry of some nups in some NPCs and need not represent a new class of specialized NPC (see also comment 4).

The reviewer is obviously correct, we cannot resolve individual NPCs in the cap. The complete absence of the Mlp proteins in the cap, however, suggests that the cap NPCs are fairly uniformly lacking them. Concerning Nup60, the discussion is more complicated because it can bind the inner nuclear membrane even when they are not located at NPCs (Meszaros et al., 2015). It is impossible at this stage to determine whether the residual signal observed for Nup60 in the cap is due to its presence at a subset of NPCs in the cap, its reduced presence in all cap NPCs or its absence from NPCs and binding to the INM between the NPCs. Nevertheless, all or a substantial fraction of the NPCs in the cap are differently composed than the bulk of NPCs, suggesting that they might be specialized. As discussed below, an alternative is that they are not specialized but simply incomplete or damaged. We now discuss this in the corresponding section.

Second, it cannot be differentiated (based on the current data) whether the NPCs in the cap are a product of nup loss or defective NPC assembly. With the latter in mind, in a model in which the cap has defectively formed NPCs, one could imagine a scenario in which they aren't formed correctly because there is simply less of a given nup produced in the mother cell (or perhaps some are targeted by recently discovered autophagy pathways that target Nup159, e.g. Tomioka et al., 2020; Lee et al., 2020). Evaluating the total levels of the Nup fusions, at least at the level of total fluorescence, could help rule out this possibility.

As discussed above, our available data suggest that the NPCs in the cap are a product of Nup loss, mediated by SAGA-dependent modification of at least Nup60. We recognize that while this is the simplest explanation, more complex hypotheses have not been excluded. For example, Mlp proteins might mediate the recruitment of circles to NPCs only indirectly via their effect on non-bound NPCs and SAGA would prevent the subsequent recruitment of Mlps to circle-bound NPCs. However, even in this case the lack of the basket is mediated by circle attachment. Furthermore, our new data with additional SINC components (Figure 4) strengthen the notion that at least the cell does not recognize these NPCs as defective.

Finally, we have evaluated the amount of total Nups fluorescence available in the old cells. As the reviewer can see in Author response image 1, the average intensity of Nup60 and Mlp1 throughout the cell remains fairly stable between young and old cells. These data are consistent with the idea that at there is at least no dramatic drop in their availability. Consistent with them not being retained like core Nups together with the circles in the mother cell, the levels of Nup60 and Mlp1 do not increase as for Nup170 for example.

**Author response image 1. sa2fig1:** 

3) SAGA is thought to be bound to many NPCs irrespective of ecDNA where it presumably acetylates Nup60 as suggested by the authors. Can the authors clarify whether it is SAGA, or the tethering of ecDNA to SAGA, that triggers the putative remodeling? As I can't think of a straightforward experiment to test this, I suggest at least discussing caveats associated with the model along this point.

As suggested by the reviewer, we would ideally need to distinguish two events. The initial removal of the basket from NPCs and the maintenance of the specialized NPC organization in the cap. Since Nup60 is a rapidly exchanging Nup (see Hakhverdyan et al., Mol. Cell 2021 and Denoth-Lippuner et al., 2014), its exclusion from the cap at steady-state is specifically maintained at the circle-bound NPCs. The fact that Nup60 can reappear at the cap upon mutation of one of its acetylation sites indicates that its continuous SAGA-dependent acetylation is needed to maintain it away from the cap. Interestingly, this observation also establishes that the circle-bound NPCs are competent for Nup60 recruitment. Thus, we have little doubt that the recruitment of SAGA to circle-bound NPCs, specifically, ensures the maintenance of their specific stoichiometry. Concerning the initial events generating these NPCs, we agree that it will be difficult with current methods to determine their precise order. However, this might actually not be biologically as relevant as the mechanisms maintaining the circle-attached NPCs in their specialized state.

4) There are several results that may be inconsistent with the literature/our understanding of NPC structure that need to be considered and/or explained: a) That Nup159 and Nup82 remain at NPCs while Nup116 is missing (Figure 1) is incongruent with work suggesting that at least Nup159 is missing from nup116Δ NPCs (see Allegretti et al., 2020).

This is a very interesting point, particularly given the fact that Hakhverdyan et al. (2021) also report that while Nup82 and Nup159 are highly stable at NPCs (exchange takes more than 7 hours), Nup116 is mobile (exchange takes place in 3-7 hours at bulk level). Thus, it is well possible that while Nup116 is required for the proper recruitment of Nup159 and Nup82 to NPCs, it becomes irrelevant for their localization once they are properly inserted in the NPC. Interestingly, the presence of Nup159 and Nup82 supports the idea that Nup116 must have been present at the corresponding circle-bound NPCs at some point before or during circle attachment. These NPCs are unlikely to be simply incomplete to start with. We now discuss this point at the end of the paper.

b) The localization of NTRs particularly Kap60 is a bit confusing as in the absence of Nup60 it localizes in the nucleus not the nuclear periphery (Denning et al., 2001) as Nup2 is also mislocalized in this strain. Thus it seems counterintuitive that Kap60 would be found in the NPC cap as suggested in Figure 7.

This apparent discrepancy could be explained again by the fact that the authors were looking at the localization of Kap60 in cells completely lacking Nup60, which is not the case in the cells we imaged. In the cells containing a circle cluster, the circle-free NPCs, which are many, recruit Nup60, allowing for Kap60 export from the nucleus and its reentry through NPCs. Also, the studies of Denning et al. were carried out by immunofluorescence, which required fixation of the cells. Fixation frequently affects the localization of dynamic proteins: Their fixation takes place where the protein is most accessible to the fixation agent. In that respect, it is remarkable that in their images Kap60 never localizes to the nuclear periphery, not even in wild type cells, while Kap60-GFP does in otherwise fully wild type cells, even those devoid of circles. Note that tagging Kap60 with GFP does not seem to affect its function since the cells relying solely on the tagged version of the protein are fully viable. The *kap60∆* cells are dead.

[Editors' note: further revisions were suggested prior to acceptance, as described below.]

Essential revisions:Reviewer 3 asks for some further clarification on SINC (see the comments below). Please address or respond to this remaining concern.Reviewer #3 (Recommendations for the authors):The authors were able to address the majority of my concerns in the revised manuscript. Some clarification is needed about the SINC, however. The SINC was first described in Webster et al., Cell, 2014 as an accumulation of misassembled NPCs that arises upon deletion of VPS4 or SNF7 and is retained in mother cells. Thus, neither Vps4 nor Snf7 localizes to the SINC as suggested by the authors (but Chm7 does, see Webster et al., 2016). Further, the idea that the SINC actively recognizes defective NPCs or that anything is "targeted by the SINC" is probably inaccurate and should be reworded to "targeted by ESCRTs". For example, there is evidence that ESCRTs like Chm7 (but also likely Snf7 and Vps4) localize to the nuclear envelope when NPC assembly is perturbed (Webster et al., 2016). Thus, these and other data have implicated ESCRTs in a form of NPC quality control. To be clear, nothing needs to change with regards to the data itself but I suggest that the authors are a bit more circumspect when labeling the Snf7/Vps4 focal nuclear envelope localization as SINCs.

We have now addressed the last point of reviewer 3 by ensuring that the text clearly distinguishes the demonstrated marker of the SINC (Chm7) as such and presents the ESCRTs subunits Vps4 and Snf7 as protein targeted to defective NPCs but not as being SINC components themselves, as requested by the reviewer. It is actually useful to make this distinction, indeed.